



# Impacts of snow assimilation on seasonal snow and meteorological forecasts for the Tibetan Plateau

Wei Li[1,2], Jie Chen[1,2], Lu Li[3], Yvan J. Orsolini[4], Yiheng Xiang[5], Retish Senan[6], Patricia de Rosnay[6]

[1]State Key Laboratory of Water Resources and Hydropower Engineering Science, Wuhan University, Wuhan, China
[2]Hubei Key Laboratory of Water System Science for Sponge City Construction, Wuhan University, Wuhan, China
[3]NORCE Norwegian Research Centre, Bjerknes Centre for Climate Research, Bergen, Norway
[4]NILU – Norwegian Institute for Air Research, Kjeller, Norway
[5]Institute of Heavy Rain, China Meteorological Administration (CMA), Wuhan, China
[6]ECMWF, Reading, UK

*Correspondence to*: Jie Chen (jiechen@whu.edu.cn)

**Abstract.** The Tibetan Plateau (TP) contains the largest amount of snow outside the polar regions and is the headwater of many major rivers in Asia. An accurate long-range (i.e., seasonal) meteorological forecast is of great importance for this region. The fifth-generation seasonal forecast system of the European Centre for Medium-Range Weather Forecasts (Seasonal System 5, in short SEAS5), is able to provide long-range meteorological forecast for the TP. However, the SEAS5 15 is produced without assimilating the Interactive Multisensor Snow and Ice Mapping System (IMS) snow data above 1500 m (hence over the TP in particular) which may affect the forecasting ability of SEAS5 over the region. To investigate the impacts of snow assimilation on the forecasting of snow, temperature and precipitation, twin ensemble reforecasts with and without snow assimilation above 1500 m over the TP were conducted for the spring and summer 2018. Significant changes occur in the springtime. Without snow assimilation, the reforecasts overestimate the snow cover and snow depth while 20 underestimate daily temperature over the TP. Compared to satellite-based estimates, the precipitation reforecasts perform better in the west TP (WTP) than in the east TP (ETP). With the snow assimilation, the reforecasts of snow cover, snow depth and temperature are consistently improved in the TP in the spring. However, the positive bias between the precipitation reforecasts and satellite observations worsens in the ETP. Compared to the experiment with no snow assimilation, the snow assimilation experiment significantly increases temperature and precipitation for the ETP and around 25 the longitude 95°E. The higher temperature after snow assimilation, in particular the cold bias reduction after initialization, can be attributed to the combined effects of a more realistic, decreased snowpack and of wind changes, providing favourable conditions for generating more precipitation. Overall, the snow assimilation can improve the seasonal forecasts by affecting the surface energy budget.

## 1 Introduction

The Tibetan Plateau (TP) is often regarded as the "Third Pole" due to high altitudes and complex terrains (Qiu, 2008), and plays an important role in the atmospheric circulation of the northern hemisphere, regulating mid-latitude westerlies and the



Asian monsoon system (Yang et al., 2014; Yang et al., 2019; Chen et al., 2020). In addition, the TP is the headwater of many major rivers in Asia, such as the Indus, Brahmaputra, Yellow, Yangtze and Lancang-Mekong River. Thus, it is also regarded as the "Asian water tower" (Immerzeel et al., 2010; Kuang and Jiao, 2016). Considering the special role of the TP, an

accurate long-range (i.e., seasonal) meteorological forecast in this region would provide a reliable meteorological background for the downstream regions, and further bring huge socioeconomical benefits through the prediction of meteorological and hydrological processes (Hansen, 2002; Shafiee-Jood et al., 2014; Clark et al., 2017; Ceglar et al., 2018; Li et al., 2019).

The fifth-generation seasonal forecast system of the European Centre for Medium-Range Weather Forecasts (ECMWF)

(Seasonal System 5, in short SEAS5) (Johnson et al., 2019) replaced its predecessor System 4 and became an operational forecast system in November 2017 (Molteni et al., 2011). The SEAS5 provides operational meteorological forecasts for a lead time of up to 7 months with an ensemble of 51 members. Reforecasts with 25 members have also been used to evaluate the ability of SEAS5 in forecasting temperature and precipitation. For example, Wang et al. (2019) compared SEAS5 with its predecessor in the Australian continent and found that a large improvement was achieved in forecasting daily maximum

temperature and precipitation, yet with little improvement in daily minimum temperature. Gubler et al. (2020) found that the SEAS5 was reliable in forecasting temperature and precipitation in many regions of South America affected by ENSO variability. In addition, Ehsan et al. (2020) showed that SEAS5 could capture the observed climatological mean and variability patterns of peak summer monsoon precipitation over Pakistan, despite being biased over complex topography zones. Chevuturi et al. (2021) indicated that SEAS5 performed well in forecasting dynamical features of the large-scale

monsoon one month ahead.

The impact of rapid snow variability over the TP during winter and spring on medium-range to subseasonal forecasts has recently been investigated by Li et al. (2018; 2020b). On the longer seasonal time scale, the impact of the snow initialization in seasonal prediction system -in particular in SEAS5- has not been evaluated, especially in the spring season. Considering the special climate and topography in this region, a first evaluation of SEAS5 forecasts for the surface fields is needed, as

well as for precipitation. More importantly, the SEAS5 forecasts were produced based on the Integrated Forecast System (IFS) atmosphere model cycle 43r1 (IFS cycle 43r1, 2016) without assimilating the Interactive Multisensor Snow and Ice Mapping System (IMS) snow data above 1500 m, including over the TP. The same restriction applies to ERA5 reanalyses and to operational ECMWF medium-range forecasts (Orsolini et al., 2019). The impact of removing IMS snow data in mountainous areas above 1500 m was detailed in De Rosnay et al. (2012). They showed that the new ECMWF snow analysis

combining improvements of the analysis approach (OI vs Cressman method) and data pre-processing and quality control (IMS snow cover product resolution and implementation of a 1500 m altitude threshold) had an overall positive impact on the atmospheric forecast skill, with root mean square error forecast for the 1000 hPa geopotential height improved by 1–4 % in the short range (forecasts until day 4). This altitude threshold has been used since its implementation in November 2010, including in the recent IFS cycles used for ERA5 (41r2) and in the current operational cycle. However, the lack of





assimilating the IMS snow data above 1500 m might influence the forecasting ability of SEAS5 over the TP (Wang et al., 2020; Lin et al., 2021), and inclusion of IMS over the Tibetan Plateau might be beneficial at the regional scale.

In order to investigate the impacts of snow assimilation over high altitude mountainous areas on the forecasting ability of the SEAS5, twin experiments with and without the IMS snow data assimilation (DA) above 1500 m over the TP were conducted as a case study in the year 2018. The configuration for these experiments was largely similar to the current SEAS5 but with

lower atmospheric (Tco199) and ocean (ORCA1_Z75) resolution and a newer IFS model cycle (CY45R1) (IFS cycle 45r1, 2018). Using these twin experiments, this case study investigates how snow assimilation over the TP influences the long-range prediction of snow, temperature and precipitation over the TP.

## 2 Study area and data

### 2.1 Study area

This study focuses on the TP within China (25-40°N, 73-105°E) (Fig. 1). In most parts of the TP, the altitude is higher than 2500 m. Precipitation is influenced by the westerlies, the South Asian and the East Asian summer monsoon systems (Schiemann et al., 2009; Yao et al., 2012; Yang et al., 2014). Specifically, precipitation in the southeastern TP is under the control of the warm and humid Indian monsoon (Li et al., 2020a), with the multiyear-averaged precipitation being more than 2000 mm, and most of the precipitation concentrated between May and September. As moisture transport is blocked by high

mountains, precipitation in northwestern TP is reduced to less than 50 mm (Curio and Scherer, 2016). In addition, the multiyear-averaged temperature changes from 20°C to below -6°C from southeast to northwest. The climate pattern in the eastern TP (ETP) is usually considered as wet, while it is usually considered as dry in the western TP (WTP) (Li et al., 2020a). Considering the huge spatial variability of the precipitation and temperature in the TP, the study area for our analysis was divided into the ETP and the WTP by the longitude 95°E.

### 2.2 Data

Twin experiments with and without the IMS snow DA above 1500 m over the TP were conducted for the spring 2018 using a coupled prediction system close to SEAS5. Both reforecasts start from April 1st, 2018 with a lead time of 4-month, i.e., from April 1st to July 31st, 2018. Each reforecast consists of 25 ensemble members. In order to analyse the seasonal changes in the reforecasts with snow DA and without, April 1st to May 31st was defined as spring, while June 1st to July 31st was

defined as summer. The reforecasts have a spatial resolution of 0.5°. The output temporal resolution ranges from 6-hour to 24-hour depending on the variable. In this study, total liquid precipitation (hereafter precipitation), 2m air temperature, snow cover fraction, snow depth, snowfall, snow density, forecast albedo and 10m wind were used for analysis. The IMS snow data used in this study was retrieved from the National Snow and Ice Data Centre (NSIDC) and has a resolution of 4 km. More details about this dataset can be found in https://nsidc.org/data/g02156. In this study, the 4-km IMS snow cover was



processed through zonal statistics to get the IMS snow cover fraction with a spatial resolution of 0.5° which was consistent with the spatial resolution of the two ensemble reforecasts.

Gridded temperature and precipitation from multiple sources were used to benchmark the ability of the twin reforecasts because of sparse meteorological stations in the TP. The gridded temperature dataset (CN05.1) was generated based on the 2416 meteorological stations in China by Wu and Gao (2013) and had been used in many other studies (Xu et al., 2009; Wu

et al., 2017). The CN05.1 temperature dataset is at the daily scale and has a spatial resolution of 0.25°. The gridded precipitation includes Global precipitation measurement (GPM), which is an international satellite mission launched by the National Aeronautics and Space Administration (NASA) and the Japanese Space Agency (JAXA) (Hou et al., 2014). The spatial and temporal resolutions of GPM are 0.1° and half-hourly, respectively. GPM has been compared with other satellite precipitation products in many studies (Guo et al., 2016; Tan and Duan, 2017; Prakash et al., 2018) and ranks top among

them. Besides the gridded data, in-situ temperature and precipitation observations in TP were also used. There are 64 meteorological stations in total, and most of them are located in the ETP. The gauged data were quality-controlled and provided by the China Meteorological Data Sharing Service System, and were also used in the generation of the CN05.1 dataset.

In addition, a daily snow cover fraction dataset for TP (hereinafter: TPSCF) provided by China National Cryosphere Desert

Data Centre was used. The dataset was produced based on MODIS normalized snow index data with the spatial resolution of 500 m, combing with the terrain data and a variety of snow cover estimation algorithm, realized re-estimation of snow cover under the conditions of cloud cover. The dataset only has data from January to June in each year. More details about this dataset can be found in https://www.scidb.cn/en/detail?_dataSetId=633694460970008576&dataSetType=journal#. Moreover, a daily snow depth dataset for TP (hereinafter: TPSD) produced by Yan et al. (2021) was also used. The TPSD dataset was

derived from the fusion of snow probability data and the long-term series of snow depth dataset over China and has a spatial resolution of 0.05°. More details about the TPSD dataset can be found in http://data.tpdc.ac.cn/zh-hans/data/0515ce19-5a69-4f86-822b-330aa11e2a28/.

## 3 Methods

The configuration of the twin experiments for this case study in the year 2018 was similar to the current SEAS5 but with

lower atmospheric (Tco199) and ocean (ORCA1_Z75) resolutions and a newer IFS model cycle (CY45R1). The ocean and sea-ice initial conditions for the twin experiments were provided by the new operational ocean analysis system OCENAN5 (Zuo et al., 2019) which was made up of the historical ocean reanalysis and the daily real-time ocean analysis. The atmospheric and land initial conditions for both experiments were obtained from dedicated analysis experiments with the ECMWF land data assimilation system (LDAS). Details about the LDAS can be found in the Dee et al. (2011) and De

Rosnay et al. (2014) or in https://www.ecmwf.int/en/elibrary/18712-ifs-documentation-cy45r1-part-ii-data-assimilation. Here, we used twin experiments that differed only in the land initial states: the control experiment (h4uy) included the



assimilation of daily, 4-km IMS snow cover below 1500 m globally, as in SEAS5, while the sensitivity experiment (h4uz, the DA experiment) included in addition assimilation of the same IMS snow cover above 1500 m over the TP.

To generate the 25-member ensemble, initial condition perturbations to atmosphere and ocean initial conditions and perturbations to the atmospheric model were applied. Perturbing the initial conditions is used to represent uncertainty in the initial state and to increase the ensemble spread. Among all members, ensemble member 0 was initialized from unperturbed atmospheric initial conditions, while in other members, all upper-air fields and a limited set of land fields (snow, soil moisture, soil temperature, skin temperature and sea-ice temperature) were perturbed. The perturbation of the atmospheric model is use to represent uncertainty from missing or unresolved sub-grid-scale processes (e.g. clouds, convection, radiation,
turbulence) which have to be parameterized (Palmer, 2012).

## 4 Results

### 4.1 Changes in snow variables with the snow assimilation

Considering that the only difference between the twin experiments is whether having IMS snow assimilation above 1500 m over the TP or not, the snow cover and snow depth are firstly analysed to evaluate the effects of the snow assimilation. The
spatial differences in snow cover fraction between IMS and TPSCF, and between the ensemble reforecasts and TPSCF in spring are presented in Fig. 2a-c. For most places of the TP, the snow cover fraction of IMS and the two reforecasts are larger than the TPSCF snow cover fraction. The differences between the IMS and TPSCF snow cover fraction (IMS minus TPSCF) are smaller than 0.4 for most places of the TP. The snow cover fraction of the control reforecasts is significantly larger than the TPSCF snow cover fraction around the boundary of the WTP and ETP where the differences (the control
reforecasts minus TPSCF) are larger than 0.6. Meanwhile, the differences in snow cover fraction when the DA reforecasts minus TPSCF are smaller than 0.4 for most places of the TP, which is consistent with the differences between IMS and TPSCF. Figure 2d-f presents the spatial differences in snow cover fraction between the two reforecasts. In both the spring and the whole period, with added snow assimilation, the snow cover fraction of the DA reforecasts is significantly smaller than that of the control reforecasts for most places of the TP, especially for the ETP and around the boundary of the WTP and ETP. However, in summer, the differences between the two reforecasts are small and range from -0.1 to 0.1 for most
places of the TP. Overall, the positive bias in snow cover is much reduced in the DA reforecasts.

Figure 3 presents the spatial differences in snow depth between the ensemble reforecasts and TPSD, and between the two reforecasts. The spatial differences in snow depth are similar with those in snow cover fraction in spring. Generally, the snow depths of the two reforecasts are larger than the TPSD snow depth for most places of the TP. However, in both the
spring and the whole period, the snow depth of the control reforecasts is significantly larger than the TPSD snow depth around the boundary of the WTP and ETP in the southern TP. The differences in snow depth between the two reforecasts (the DA reforecasts minus the control reforecasts) range from -60 cm to 6 cm. The positive bias in snow depth is also much reduced in the DA reforecasts, which is consistent with the assimilation of IMS snow cover removing snow in regions where



the inherent model precipitation excess builds an unrealistic snowpack. The snow depth of the DA reforecasts is less than

that of the control reforecasts at the 5% significance level for most places of the TP, especially for the ETP and around the

boundary of the WTP and ETP in the southern TP. As for summer, the spatial distributions of snow depth are similar

between the two reforecasts. The differences between the two reforecasts range from -6 to 6 cm for most places of the TP.

Considering that the snow cover and snow depth of the reforecasts change significantly after snow assimilation, the spatial

differences in snowfall and snow density between the two ensemble reforecasts are further analysed (Fig. 4). In either the

spring or the whole period, the snowfall of the DA reforecasts is more than that of the control reforecasts in the southeastern

TP, especially around the boundary of the WTP and ETP, while the results are reversed in the WTP. Moreover, the

differences in snowfall between the two reforecasts (the DA reforecasts minus the control reforecasts) range from -0.2 to 0.8

mm of water equivalent, and the spatial differences are statistically significant at the 5% significance level mainly for

regions where the differences (the DA reforecasts minus the control reforecasts) are larger than 0.3 mm of water equivalent.

In summer, the snowfall of the DA reforecasts is more than that of the control reforecasts in the southwestern TP, while the

results are reversed in the northeastern TP. The differences in snowfall between the two reforecasts range from -0.2 to 0.2

mm of water equivalent for most places of the TP.

As for the snow density, in either the spring or the whole period, the snow density of the DA reforecasts is smaller than that

of the control reforecasts for most places of the TP, especially for the ETP and around the boundary of the WTP and ETP in

the southern TP, while the results are reversed in the southwestern TP. The spatial differences in snow density between the

two reforecasts are statistically significant at the 5% significance level for regions where the absolute differences are larger

than 25 kg/m$^3$. In summer, the snow density of the DA reforecasts is larger than that of the control reforecasts in the eastern

and southwestern TP and the differences between the two reforecasts are smaller than 25 kg/m$^3$.

Since the changes in snowfall and density leads to changes in surface albedo after snow assimilation, Figure 5 presents the

spatial differences in forecast albedo between the two ensemble reforecasts. In either the spring or the whole period, the

forecast albedo of the DA reforecasts is smaller than that of the control reforecasts for most places of the TP, especially for

the ETP and around the boundary of the WTP and ETP in the southern TP. The differences in forecast albedo between the

two reforecasts (the DA reforecasts minus the control reforecasts) range from -0.2 to 0.04 for most places of the TP. The

significant differences in forecast albedo between the two reforecasts are mainly observed in regions where the absolute

differences are larger than 0.04. However, in summer, the forecast albedo of the DA reforecasts is larger in the southern TP

while smaller in the northern TP comparing with that of the control reforecasts. The differences in forecast albedo after snow

assimilation range from -0.04 to 0.04 for most places of the TP.

In summary, the main points are that snow assimilation reduces the positive snow bias in the spring over most of the TP,

while its impact is limited in the summer, and all the snow variables changes significantly after snow assimilation for the

ETP and around the boundary of the WTP and ETP in the southern TP. The reduced snowpack leads to a diminished surface

albedo.





### 4.2 Evaluation of the temperature and wind reforecasts

#### 4.2.1 Evaluation of the temperature reforecasts

Figure 6 presents the temperature time series from April 1st to July 31st for the two ensemble reforecasts and CN05.1 data.
The time series was smoothed by the 5-day moving window. The blue area and line represent the ranges and ensemble-mean of the control reforecasts, while the orange area and line represent the ranges and ensemble-mean of the DA reforecasts, and the black line represents CN05.1 data. In the WTP (Fig. 6a), the ensemble-means of the temperature reforecasts are lower than the CN05.1 temperature. However, the DA reforecasts are in excellent agreement with the CN05.1 temperature at the initial time (thereby reducing the large initial bias as expected from a reduced snowpack) and are slightly closer to CN05.1 in the first month and a half. In the ETP (Fig. 6b), the initial bias reduction is even larger (about 5 K) and, while the ensemble-means of temperature reforecasts are lower than the CN05.1 temperature for most of the time, it remains closer to the CN05.1 temperature for about one month and a half. The temperatures show little change between both reforecast ensemble-means after June, consistent with the lack of change in the snowpack in summer.

The basin-averaged CCs and mean absolute error (MAEs) of daily temperature between the two ensemble reforecasts and CN05.1 data are presented in Fig. 7. In the WTP, the CCs of temperature are higher than 0.80. The median and mean values of the CCs are smaller after snow assimilation. The MAEs between the temperature reforecasts and CN05.1 temperature are smaller than 4.2 ℃ and become smaller after snow assimilation. In the ETP, the CCs are higher than 0.78 and the MAES are smaller than 3.6 ℃. As for the WTP, the MAEs of the DA reforecasts are smaller than those of the control reforecasts, indicating that the snow assimilation improves the temperature forecasts. Furthermore, the correlations and mean error of daily temperature between the temperature reforecasts and CN05.1 temperature are lower in the ETP than in the WTP. In Fig. S1, the daily temperature of the two ensemble reforecasts is compared with that of the CN05.1 data. In general, the temperature reforecasts are lower than the CN05.1 temperature. However, the temperature of the DA reforecasts is closer to the CN05.1 temperature, consistent with the high snow bias reduction.

Figure 8 presents the spatial differences in daily temperature between the ensemble reforecasts and CN05.1 data, and between the two reforecasts. In spring, the temperature reforecasts are lower than the CN05.1 temperature for most places of the TP. After snow assimilation, the reforecasts become closer to the CN05.1 temperature, especially for the ETP and around the boundary of the WTP and ETP. In summer, for most places of the TP, the temperature reforecasts are lower than the CN05.1 temperature and the spatial differences in daily temperature between the two reforecasts range from -0.4 ℃ to 0.4 ℃. For the whole period, the spatial differences between the temperature reforecasts and CN05.1 daily temperature are similar to those in spring. The most distinctly spatial characteristic is that the temperature of the DA reforecasts is significantly higher than that of the control reforecasts for the ETP and around the boundary of the WTP and ETP. Moreover, the spatial differences between the temperature reforecasts and CN05.1 daily temperature are statistically significant at the 5% significance level for places where the absolute differences are larger than 2.0 ℃, while the statistically significant regions of



the spatial differences between the two reforecasts are mainly concentrated in regions where the differences are larger than 1.2 ℃.

The spatial correlations and mean error between the temperature reforecasts and in-situ observations are displayed in Fig. S2. The temperature reforecasts are interpolated to stations using the bilinear interpolation method. The results show that with no snow assimilation, the spatial correlations between the temperature reforecasts and in-situ observations are 0.64 and 0.66 for spring and the whole period, respectively, while with snow assimilation the spatial correlations are 0.68 for both spring and the whole period. Moreover, the mean error is 7.26 ℃ in spring, 4.99 ℃ in summer and 6.12 ℃ in the whole period before snow assimilation, while the mean error is 5.71 ℃ in spring, 4.88 ℃ in summer and 5.30 in the whole period after snow assimilation. In general, the spatial correlations are higher, and the mean error is smaller when performing snow assimilation.

### 4.2.2 Changes in wind field and geopotential height with the snow assimilation

It is noticeable that the significant differences in snow variables between the two reforecasts, while present over most of the TP in the spring, nevertheless maximize for the ETP and around the boundary of the WTP and ETP in the southern TP (Fig. 2-5), which is consistent with the spatial changes in temperature (Fig. 8). Furthermore, besides the local impacts of snow assimilation on temperature, the horizontal heat transfer is also influenced by the snow assimilation. Therefore, the changes in 10 m horizontal wind field after snow assimilation are also analysed (Fig. 9). With snow assimilation, there is an obvious centre of the changes in wind field in the ETP in either the spring or the whole period. The wind speed of the DA reforecasts is much larger than that of the control reforecasts in the ETP. Moreover, the closer to the centre of the ETP, the larger the wind speed increase. However, the snow assimilation has little impact on the 10 m wind field in summer.

The geopotential height at 600 hpa is also presented to analyse the cyclonic anomalies with added snow assimilation (Fig. 10). In spring or the whole period, the geopotential height at 600 hpa of the DA reforecasts is lower than that of the control reforecasts for the whole TP, especially for the ETP and around the boundary of the WTP and ETP. The significant differences in geopotential height at 600hpa between the two reforecasts are mainly observed in regions where the absolute differences are larger than 4 gpm. The results are consistent with convergence and ascent, and are also consistent with past results in previous study: Zhang et al. (2021) found cyclonic anomaly over TP, i.e., increased low-level convergence and ascent, was in response to decreased snow cover in late spring. However, in summer, the geopotential height at 600 hpa of the DA reforecasts is higher than that of the control reforecasts over most of the central and eastern TP, while the results are reversed in the western TP.

### 4.3 Evaluation of the precipitation reforecasts

Despite the notable improvements in the predicting snow and surface temperature in the snow assimilation forecasts, at least in the first month and a half, it remains to be seen if these translate to precipitation. Actual predictability studies with dynamical prediction systems stressed that a more realistic land initialization improves surface temperature forecasts but the impact on precipitation remains weaker (Koster et al., 2010; 2011). Figure 11 presents the precipitation time series from



April 1st to July 31st for the two ensemble reforecasts and the GPM data. As for temperature, the time series was smoothed by using the 5-day moving window and the black line represents GPM data. In the WTP, the ensemble-mean precipitation for the two ensemble reforecasts generally have the same seasonal tendency as the observations, albeit the weekly variability is smaller. There is no obvious difference in ensemble-mean precipitation between the two reforecasts. However, in the ETP,

the ensemble-mean precipitation of the DA reforecasts is higher than that of the control reforecasts, especially during a few episodes occurring mostly before June 1st. This increase could hence be related to the snow changes which were most pronounced over ETP in the spring. Moreover, the ensemble-mean precipitation of the two reforecasts is much more than GPM precipitation before June 25th, in line with the aforementioned model excess in precipitation. Although the ranges of two reforecasts are similar, those of the control reforecasts cover the GPM data better in both the WTP and ETP. However,

the upper limits of the ranges of the DA reforecasts are pretty high around June 3rd while the GPM precipitation is small.

The Spearman's correlation coefficients (CCs) and mean absolute relative error (MAREs) of daily precipitation between the two ensemble reforecasts and GPM data are presented in Fig. 12. It can be noticed that the median and mean values of the CCs between the precipitation reforecasts and GPM precipitation become smaller after snow assimilation, especially for the ETP. In other words, the correlations between the precipitation reforecasts and GPM precipitation become weak after snow

assimilation. However, the median and mean values of the MAREs become smaller after snow assimilation in the WTP, while the results are reversed in the ETP. The changes in the median and mean values of the MAREs are also larger in the ETP than in the WTP. In general, the temporal correlations are lower but the relative error is larger in the ETP than in the WTP. Furthermore, the variation ranges of the CCs and MAREs are larger for the DA reforecasts than for the control reforecasts.

Figure S3 shows the daily precipitation of the two ensemble reforecasts and GPM data. Generally, the results are quite different in the WTP and ETP. In the WTP, the daily precipitation is slightly more than the GPM data in spring for more than half members of the reforecasts. In summer and the whole period, the daily precipitation of the two reforecasts is less than that of the GPM data. Overall, the precipitation of the DA reforecasts is closer to the GPM precipitation in the WTP. In the ETP, the daily precipitation of the two reforecasts is obviously more than the GPM data, except for some members with

snow assimilation in summer. The precipitation of the control reforecasts is closer to the GPM in the ETP which is different for the situation in the WTP. In addition, the median values of the daily precipitation for the reforecasts become larger after snow assimilation.

Figure S4 presents the precipitation diurnal cycles for the two ensemble reforecasts and GPM data. In the WTP, the two reforecasts perform well in forecasting the occurrence time of peak values and the precipitation amount in spring (Fig. S4a).

The GPM precipitation diurnal cycle is absolutely covered by the envelope of the control reforecasts. In summer and the whole period, the precipitation diurnal cycles for two reforecasts are quite similar and the envelope of reforecasts can only cover the GPM precipitation amount at 0600 and 1200 UTC and they miss the 1800 UTC peak in summer precipitation. In the ETP, the ensemble means of the DA reforecasts are larger than those of the control reforecasts and the ensemble-mean precipitation of both reforecasts is more than the GPM data in spring (Fig. S4d). Only the envelope of the control reforecasts



can cover the GPM precipitation amount at 0000 and 1800 UTC. In summer and the whole period, the precipitation diurnal cycles of the reforecasts are considerably biased compared with the diurnal cycles for GPM precipitation. For most members of the reforecasts, the precipitation reforecasts show a sharp peak at 1200 UTC and exceed the GPM precipitation at 0600 and 1200 UTC while lower at 0000 and 1800 UTC. In other words, the forecast model tends to systematically have a sharp summertime precipitation peak in late afternoon, which is not matched by GPM satellite observations.

The spatial differences in daily precipitation between the ensemble reforecasts and GPM data, and between the two reforecasts are displayed in Fig. 13. In spring, the reforecasts underestimate daily precipitation in the ETP while they overestimate daily precipitation in the WTP. The precipitation of the DA reforecasts is more than that of the control reforecasts in the southeastern TP, especially around the boundary of the WTP and ETP. In summer, the spatial distributions for two reforecasts are quite similar. The two ensemble reforecasts underestimate daily precipitation in the central TP while 300 overestimate daily precipitation in other regions, especially in the southern TP. Moreover, the precipitation of the DA reforecasts is more abundant than that of the control reforecasts in the WTP while less in the ETP. As for the whole period, the spatial differences between the precipitation reforecasts and GPM daily precipitation are similar to those in spring. The most significantly spatial characteristic is that the precipitation of the DA reforecasts is obviously more than that of the control reforecasts around the boundary of the WTP and ETP in the southern TP. In addition, the spatial differences between 305 the precipitation reforecasts and GPM daily precipitation are statistically significant at the 5% significance level over the whole TP, while those between two reforecasts are only statistically significant for regions where the differences are larger than 0.3 mm.

The spatial correlations and mean error between the precipitation reforecasts and in-situ observations are also presented to evaluate the ability of two reforecasts (Fig. S5). The gridded reforecasts are interpolated to stations for this evaluation using 310 the bilinear interpolation method. In general, the mean error between the reforecasts and observations is larger for the snow assimilation experiments. The spatial correlations between the reforecasts and observations become larger in spring while keeping almost unchanged for other periods after snow assimilation.

**5 Discussion**

Twin reforecasts with and without snow assimilation above 1500 m over the TP were conducted in this case study in spring 315 2018 to investigate how snow assimilation influences the long-range prediction of snow, temperature and precipitation over the TP. In general, the two ensemble reforecasts can capture the seasonal tendencies of the observed temperature and precipitation. For temperature, the temporal correlations of the reforecasts are higher than 0.78 over the TP when compared with the CN05.1 temperature. Usually, it is difficult to have high correlations in seasonal forecasting, here the results probably come from the marked seasonal cycle. However, the reforecasts tend to underestimate daily temperature. When 320 using the GPM precipitation as a benchmark, the precipitation reforecasts perform better in the WTP than in the ETP, with higher temporal correlations and smaller mean error in the WTP. Moreover, the two ensemble reforecasts usually



underestimate daily precipitation in the WTP while overestimate daily precipitation in the ETP and can better forecasts the precipitation diurnal cycles in the WTP.

The temperature reforecasts improve with the snow assimilation when comparing with the CN05.1 data. Moreover, the
temperature of the DA reforecasts is considerably higher than that of the control reforecasts, especially for the ETP and around the boundary of the WTP and ETP. With the snow assimilation, the biases between the precipitation reforecasts and GPM precipitation become larger in the ETP while smaller in the WTP, and the temporal correlations between the precipitation reforecasts and GPM precipitation become smaller. In addition, the precipitation of the DA reforecasts is significantly more than that of the control reforecasts in the southeastern TP, especially around the boundary of the WTP and
ETP, similar to the spatial changes in temperature reforecasts. The larger biases of the precipitation reforecasts in the ETP after snow assimilation may be caused by the uncertainties in observations. The bulk of the precipitation over the TP falls as snow in winter and spring, but the GPM products usually perform not well in detecting snowfall (Behrangi et al., 2014; Immerzeel et al., 2015). Besides, the in-situ stations in this region are limited and mainly located in the valleys, which may result in underestimation of precipitation (Li et al., 2021).

To better investigate how the snow assimilation above 1500 m over the TP influences the ability for temperature and precipitation forecasts, five snow variables (i.e., snow cover, snow depth, snowfall, snow density and forecast albedo) were analysed. The results indicate that the snow cover and snow depth of the two ensemble reforecasts are larger than the observations, i.e., the TPSCF snow cover fraction and TPSD snow depth, for most places of the TP. However, the snow cover and snow depth of the DA reforecasts are smaller than that of the control reforecasts for most places of the TP,
especially for the ETP and around the boundary of the WTP and ETP, which means that the snow cover and snow depth for the DA experiment are closer to the in-situ observations. Moreover, in regions where the snow cover and snow depth are smaller, the temperature is higher. The decreased snowpack of the DA reforecasts mean that less heat is required for snowmelt (Datt et al., 2008; Duffy and Bennartz, 2018). Therefore, less heat is absorbed by the snow while more heat is absorbed by the earth, leading to the higher temperature.

In addition, because of the more realistic, decreased snowpack for the DA experiment, the forecast albedo is also smaller for most places of the TP, especially for the ETP and around the boundary of the WTP and ETP in the southern TP. Therefore, more energy is absorbed by the earth due to the smaller forecast albedo, further leading to the higher temperature. With the higher temperature, the evaporation intensity becomes higher and more moisture is carried to the atmosphere (Zhang et al., 2019; Yong et al., 2021), which provide conditions for the more precipitation. Therefore, the spatial changes in precipitation
after snow assimilation are similar with those in air temperature, i.e., the precipitation of the DA reforecasts is more than that of the control reforecasts for the ETP and around the boundary of the WTP and ETP in the southern TP. Furthermore, the snowfall of the DA reforecasts is also more than that of the control reforecasts for the ETP and around the boundary of the WTP and ETP in the southern TP, which is consistent with the spatial changes in precipitation. However, it can be noticed that the largest differences in snowfall between the two reforecasts (the DA reforecasts minus the control reforecasts) reach



0.8 mm of water equivalent, while those in precipitation reach 1.8 mm, meaning that most of the increased precipitation is in the form of rainfall.

The analyses of snow variables only explain the local impacts of snow assimilation on temperature, while ignoring the horizontal heat transfer. Therefore, the 10 m wind field is also analysed and an obvious centre of changes is observed in the ETP, which is coincident with the centre of changes in snow, temperature and precipitation in the ETP, especially the closer

to the centre, the wind speed increases more. This means, with snow assimilation, the higher wind speed transfers more heat from surrounding regions to the centre. Therefore, the spatial changes show that the closer to the centre of changes in wind field, the higher the temperature rises. The higher temperature further leads to the more precipitation in the ETP for the snow assimilation experiment in the spring.

Although a comprehensive assessment of the impacts of added snow assimilation above 1500 m over the TP on the long-

range prediction of snow, temperature and precipitation was conducted, some issues remain. For example, the impacts of snow assimilation on the circulation (including upper-air) on the subseasonal-to-seasonal time scale, i.e., on the subtropical jet and downstream wave train and monsoon development remains to be investigated. This study focuses on surface level and explore how the snow assimilation influences snow, temperature and precipitation predictions through the relations among snow, temperature, wind and precipitation. Future studies will be done on pressure levels and further investigate the

impacts of snow assimilation on the circulation. Moreover, bias-correction methods (e.g., quantile mapping) are usually applied to improve temperature and precipitation predictions (Themeßl et al., 2011; Chen et al., 2013). As this study puts more emphasis on the impacts of snow assimilation, bias-correction methods can be considered in future studies to further improve the skill of seasonal forecasts.

## 6 Conclusions

Twin reforecasting experiments for the spring and summer 2018 with IMS snow DA below 1500 m globally while the other had in addition IMS snow DA above 1500 m over the TP, were used to investigate the impacts of snow assimilation on seasonal snow and meteorological forecasts over the TP. The main conclusions can be drawn as follows:

(1) The snow cover and snow depth of the two ensemble reforecasts are larger than the observations, i.e., TPSCF snow cover fraction and TPSD snow depth, for most places of the TP. The two ensemble reforecasts can capture the seasonal tendencies

of the observed temperature and precipitation, when using the CN05.1 temperature and GPM precipitation as benchmarks. For temperature, the reforecasts tend to underestimate daily temperature. The precipitation reforecasts perform better in the WTP than in the ETP even for precipitation diurnal cycles. Moreover, the reforecasts usually underestimate daily precipitation in the WTP while overestimate daily precipitation in the ETP.

(2) With the snow assimilation, the snow cover and snow depth of the reforecasts are closer to the observations. The

temperature reforecasts improve with the snow assimilation when comparing with the CN05.1 data. The biases between the precipitation reforecasts and GPM precipitation becomes larger in the ETP while smaller in the WTP, which may be partly

because of the uncertainty from the GPM observations. The snow cover and snow depth of the DA reforecasts are significantly smaller while the temperature and precipitation of the DA reforecasts are significantly larger than those of the control reforecasts for the ETP and around the boundary of the WTP and ETP.

(3) Compared to the control reforecasts, all the snow cover, snow depth and forecast albedo are smaller for the ETP and around the boundary of the WTP and ETP when performing the snow assimilation, leading to the higher temperature around these regions. The higher temperature enables more moisture to be carried to the atmosphere, providing conditions for the more precipitation. Besides, the 10 m wind field also changes in the ETP after snow assimilation, transferring more heat from surrounding regions to the centre in the ETP. The closer to the centre, the higher the temperature rises, and the higher

temperature further leads to the more precipitation in the ETP. Moreover, most of the increased precipitation is in the form of rainfall.

## Author contributions

YO, RS and PR designed the experiments and carried them out. WL prepared the manuscript with contributions from all co-authors. YX provided the station data. All co-authors participated in the analyses.

## Competing interests

The authors declare that they have no conflict of interest.

## Acknowledgements

This work has been partially supported by the Natural Science Foundation of China (Grant No. 52079093), the Hubei Provincial Natural Science Foundation of China (Grant No. 2020CFA100), the Overseas Expertise Introduction Project for

Discipline Innovation (111 Project) funded by the Ministry of Education and State Administration of Foreign Experts Affairs, P. R. China (Grant No. B18037), and the Fast-Track Initiative (FTI) project of "Contributions to the CORDEX Flaghip Pilot Study over the Tibetan Plateau Region" funded by the centre for climate dynamics (SKD) at the Bjerknes Centre for Climate Research (BCCR). We thank Professor Xuejie Gao for providing the CN05.1 gridded daily meteorological forcing dataset. YOR, RS and PdR also acknowledge the support of the International Space Science Institute in Beijing, through the working

team "Snow reanalysis over the Himalaya-Tibetan Plateau region and the monsoons" over the years 2016-2018 (Team leaders: Yvan Orsolini and Gianpaolo Balsamo).



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



# Figures

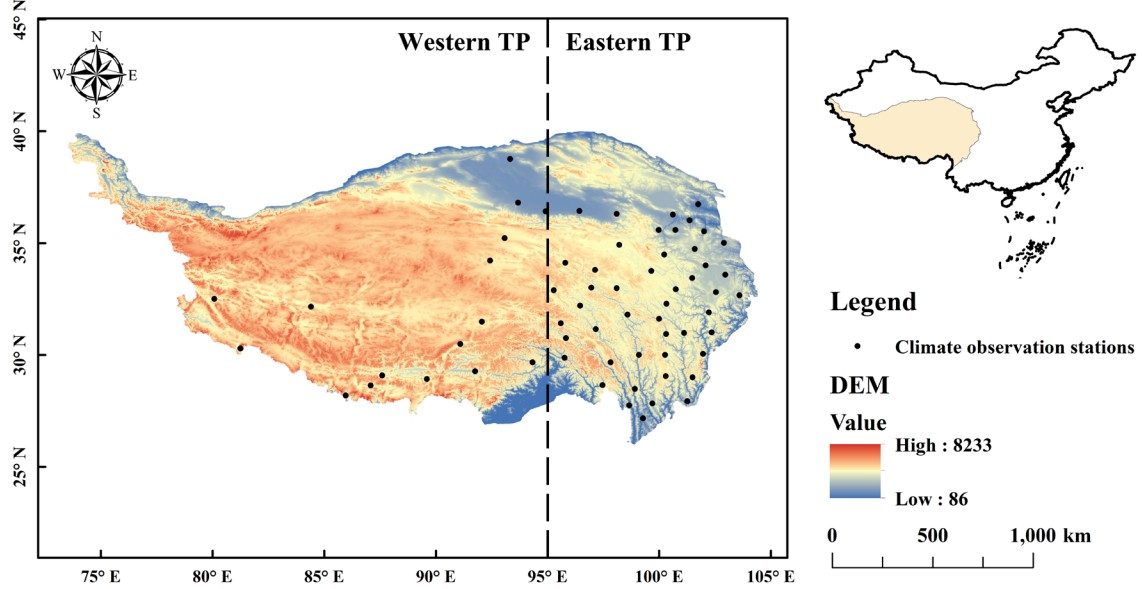


**Figure 1: The location and topography of the Tibetan Plateau and the location of climate observation stations.**





**Figure 2: (a-c) The spatial differences in snow cover fraction between IMS and TPSCF, and between the ensemble reforecasts and TPSCF in spring; (d-f) The spatial differences in snow cover fraction between the two reforecasts. The stippled regions show the statistical significance of the differences identified by the t-test at a 5% significance level.**

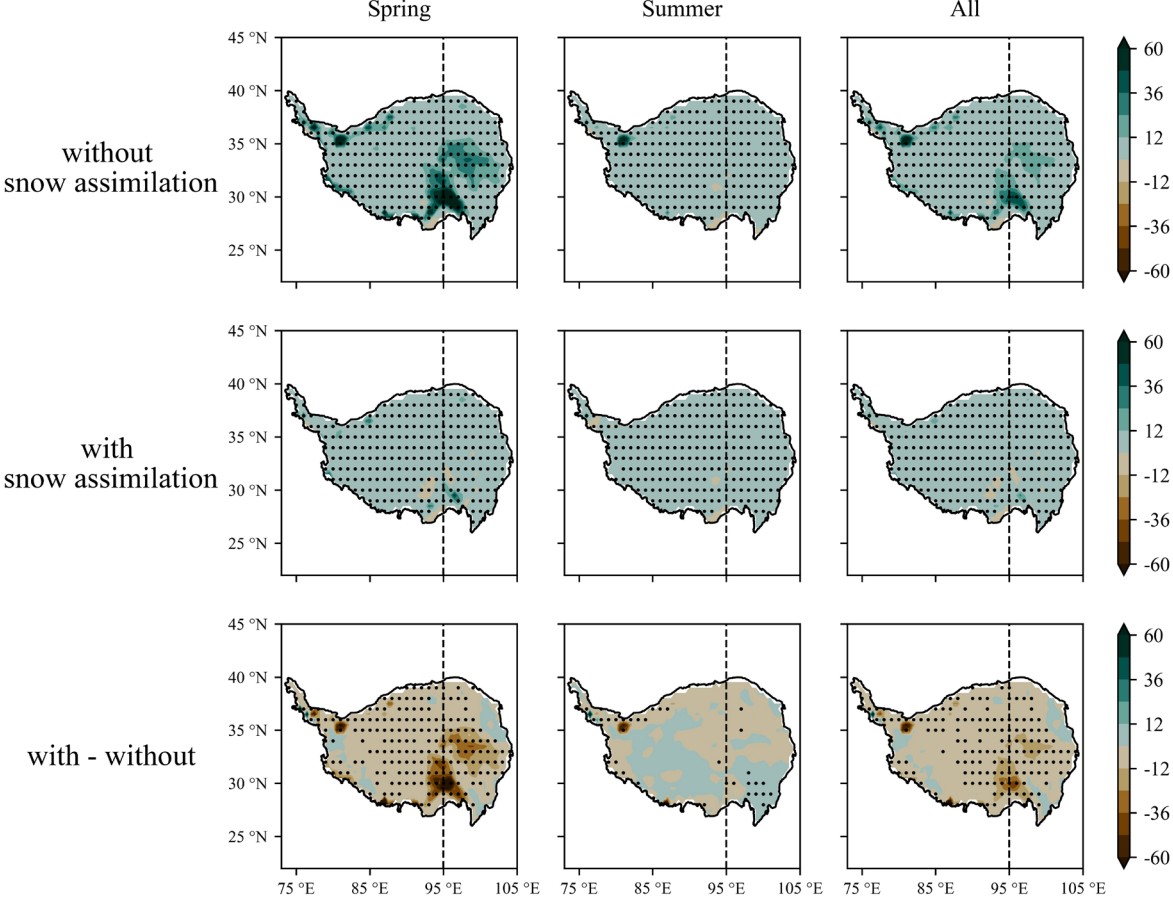

**Figure 3: The spatial differences in snow depth (cm) between the ensemble reforecasts and TPSD (left and middle columns), and**
**between the two reforecasts (right column). The stippled regions show the statistical significance of the differences identified by the**
**t-test at a 5% significance level.**


**Figure 4: The spatial differences in snowfall (mm of water equivalent) and snow density (kg/m3) between the two ensemble reforecasts. The stippled regions show the statistical significance of the differences identified by the t-test at a 5% significance level.**





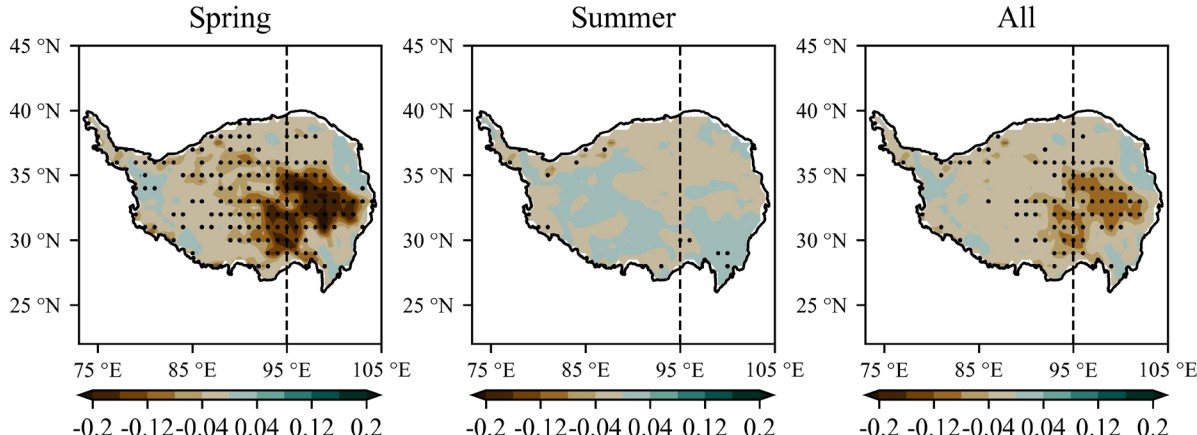

**Figure 5: The spatial differences in forecast albedo between the two ensemble reforecasts. The stippled regions show the statistical significance of the differences identified by the t-test at a 5% significance level.**






**Figure 6:** The temperature time series from April 1st to July 31st for the two ensemble reforecasts and CN05.1 data in the (a) west Tibetan Plateau and (b) east Tibetan Plateau.





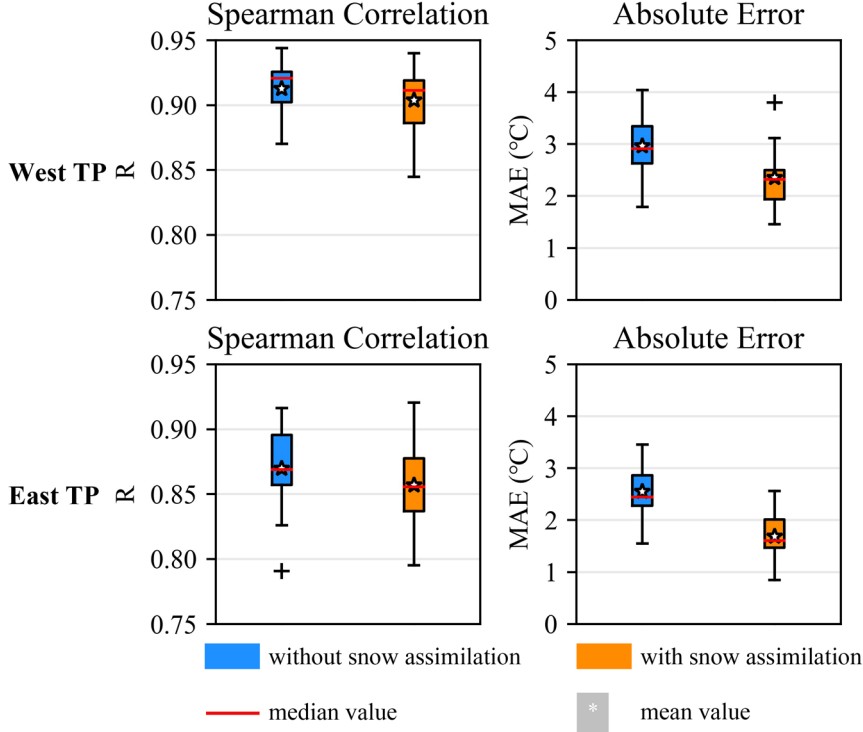

**Figure 7: The Spearman's correlation coefficient and mean absolute error of daily temperature between the two ensemble reforecasts and CN05.1 data.**

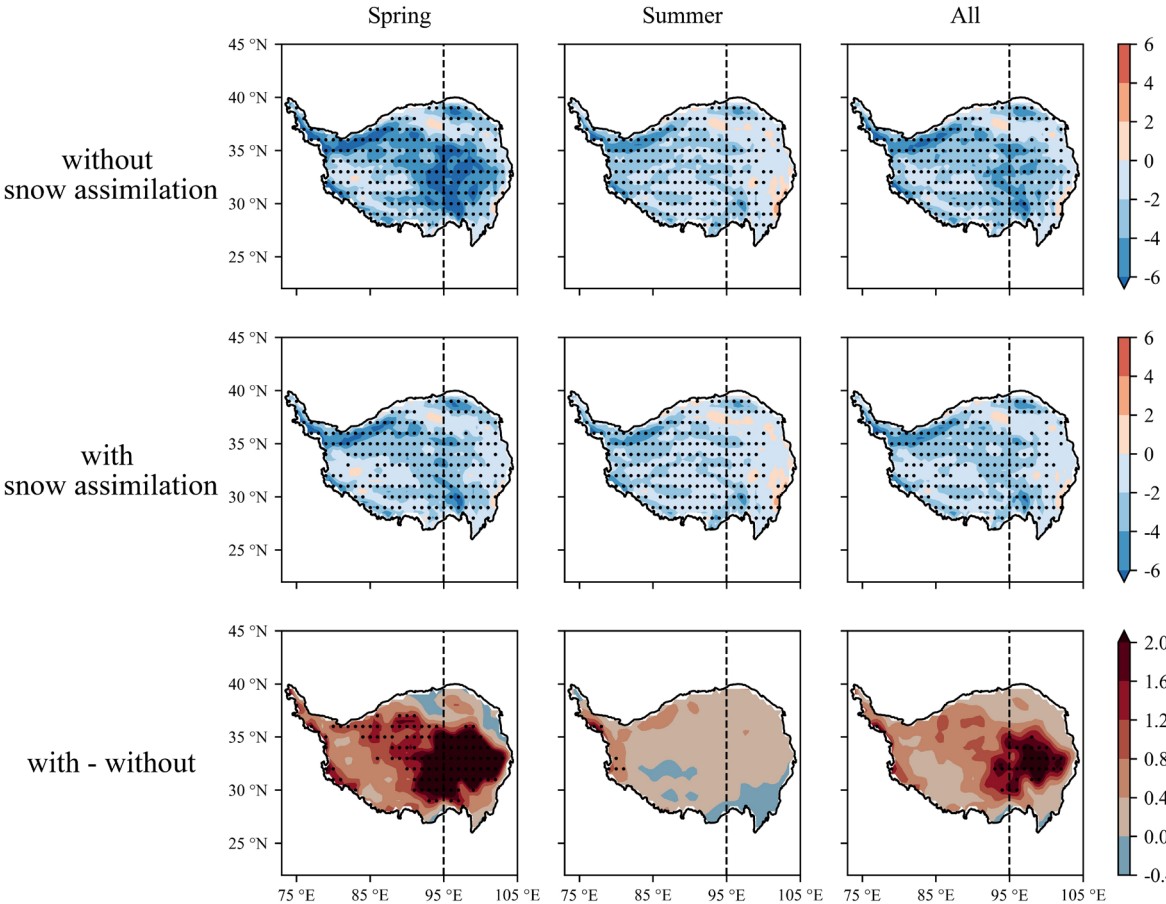

Figure 8: The spatial differences in daily temperature (°C) between the ensemble reforecasts and CN05.1 data (left and middle columns), and between the two reforecasts (right column). The stippled regions show the statistical significance of the differences identified by the t-test at a 5% significance level.





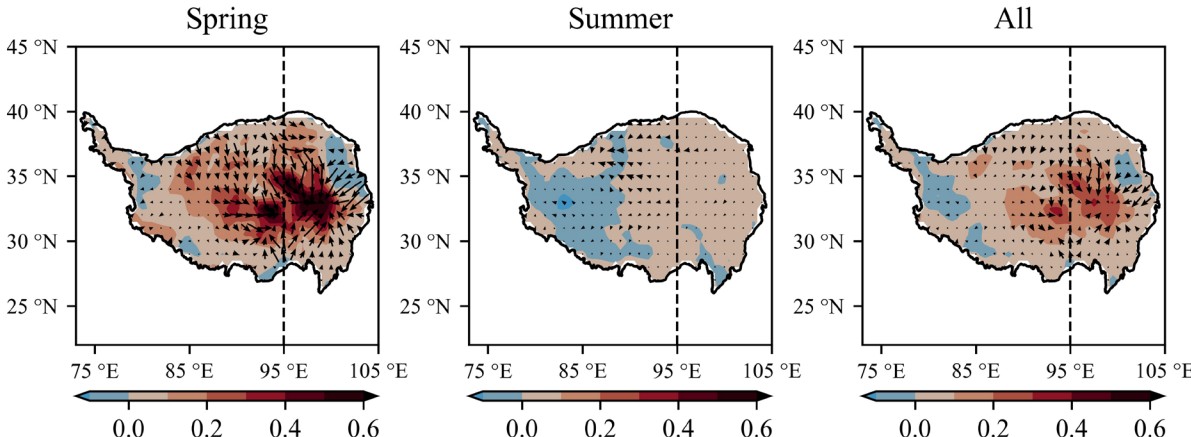

**Figure 9: The spatial differences in 10 m horizontal wind field (m/s) between the two ensemble reforecasts. The contours are wind speed.**



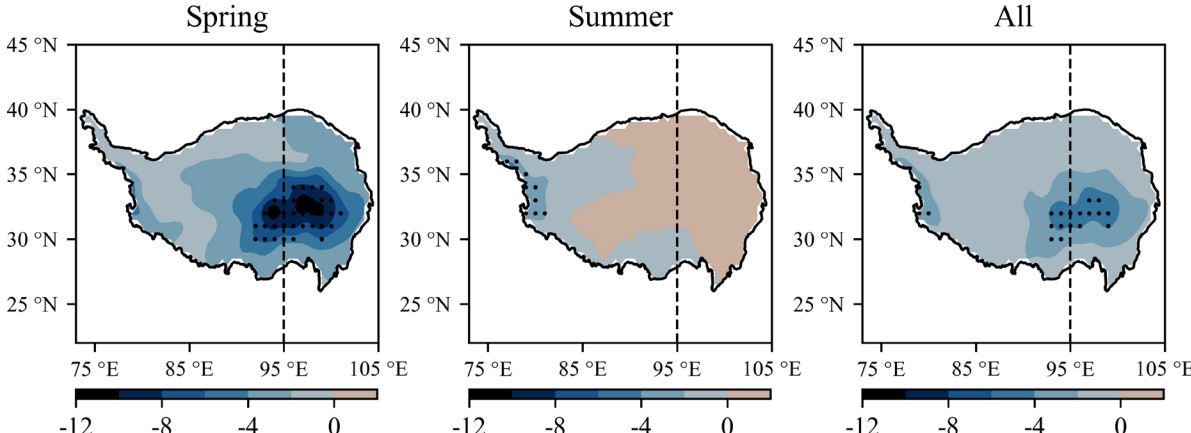

**Figure 10: The spatial differences in geopotential height at 600 hpa (geopotential meter, gpm) between the two ensemble**
**reforecasts. The stippled regions show the statistical significance of the differences identified by the t-test at a 5% significance level.**



**Figure 11: The precipitation time series from April 1st to July 31st for the two ensemble reforecasts and the GPM data in the (a) west Tibetan Plateau and (b) east Tibetan Plateau.**






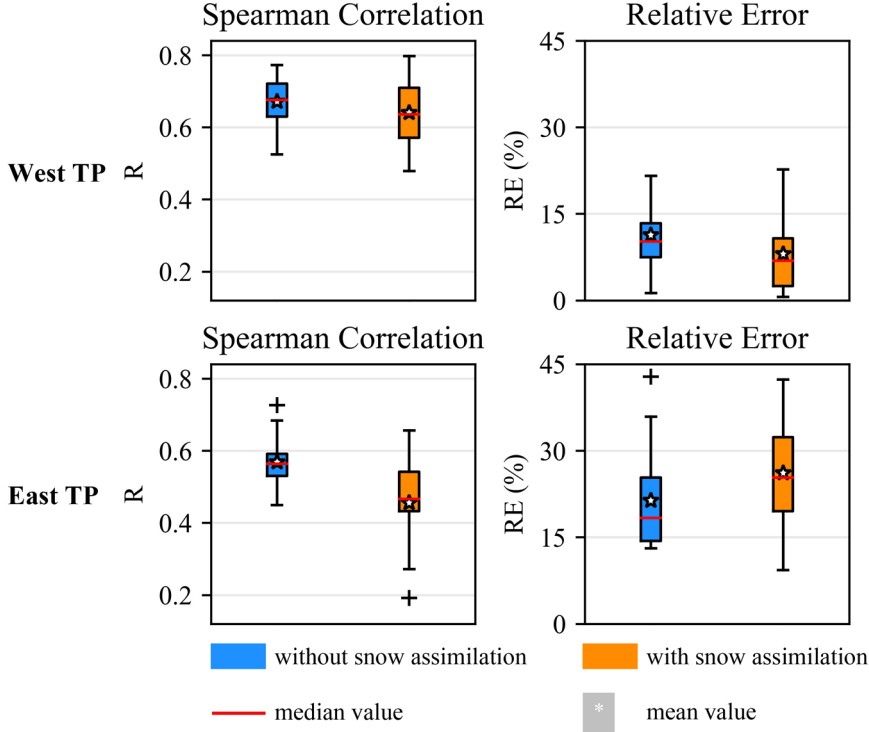

**Figure 12: The Spearman's correlation coefficients and mean absolute relative error of daily precipitation between the two ensemble reforecasts and GPM data.**




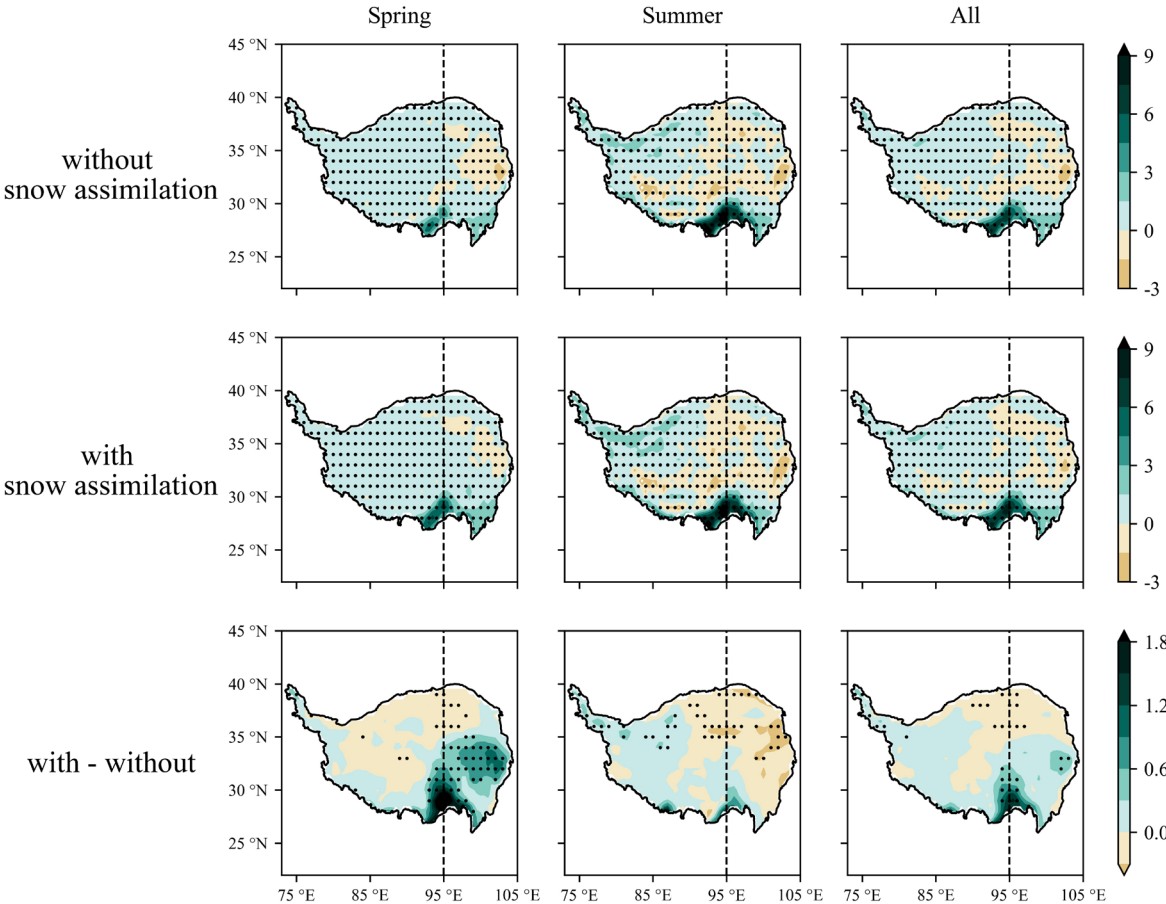

**Figure 13:** The spatial differences in daily precipitation (mm) between the ensemble reforecasts and GPM data (left and middle columns), and between the two reforecasts (right column). The stippled regions show the statistical significance of the differences identified by the t-test at a 5% significance level.