# Peer review of "Impacts of snow assimilation on seasonal snow and meteorological forecasts for the Tibetan Plateau"

_The Cryosphere, 2022_

## Author Comment (AC1)

**Reply to Referee comment 1**

Dear Editor and Reviewers:

We would like to thank the editor and all reviewers for their valuable suggestions and comments on the manuscript. These comments have not only improved the quality of the current manuscript, but also are beneficial to our future research in general. All point-by-point responses are presented as follows and we have carefully revised the manuscript based on these comments. For clarity, all comments are given in the original version, while responses are marked in blue.

**General comments:**

The manuscript describes the impact of assimilating IMS snow cover extent data over the Tibetan Plateau (i.e. over 1500 m in altitude) on the snowpack state as well as on near surface variables (temperature, precipitation, wind) and upper air variable (600 hPa geopotential height). It is therefore well within the scope of The Cryosphere.

Although the tools, models and datasets used in this paper do not bring novelty to the scientific community, the approach and goal of the study aim to provide substantial progress and answers to the community. Indeed, this study is very promising and the results bring very important information for scientists around the world.

However, the scientific method is not clearly outlined; this manuscript lacks precision leading to multiple gaps with respect to crucial details. For example, no information is given within the manuscript as to how the assimilation of IMS data is performed and very little information is given on the simulation setup leading to important doubts on the nature of the models used in this study. Furthermore, not enough details are given to describe and explain what is in the figures.

Because this study is very interesting but lacks precision, I suggest major revisions on the manuscript.

Reply: We would like to appreciate the reviewer's valuable comments on this paper. However, we beg to differ with the reviewer about the novelty of the simulations: while the underlying coupled model is based on an operational seasonal forecast model, we present unique, dedicated simulations which had not been carried out before. More information about how the assimilation of IMS data is performed and the simulation setup has been added in the manuscript, and more details about the figures have been described and explained. All the comments have been addressed and incorporated into the revised manuscript.

**Specific comments:**

Lines 55 to 57: Is SEAS5 a 3D model then? The wording "based on" confuses the readers to what is the link between SEAS5 and IFS. Please clarify with great precision, as it is crucial for the readers to have a clear idea of the tools and models you use for this study.

Reply: SEAS5 is a forecast model configuration of ECMWF Integrated Forecasting System (IFS) comprising the IFS
35   atmosphere model coupled to the NEMO 3.4 ocean model and LIM2 sea-ice model. The redundant information and
reference to IFS model cycle of SEAS5 has been removed and the above information on the relation between SEAS5 and
IFS is included in the text when SEAS5 is first described. Note also that a comprehensive description of SEAS5 is provided
by Johnson et al (2019).

40   Lines 94-96: Could you add more information about the upscaling method you used? Since IMS is a binary product (0 or 1),
how do you obtain a snow cover fraction? What assumptions have you made?

Reply: Thanks for the comment. The raw IMS snow data used in this study has a resolution of 4 km while the resolution of
the reforecasts is 0.5°. The raw IMS snow data is post-processed as following steps to get the IMS snow cover fractions with
45   the same grids of the reforecasts. Firstly, the raw IMS snow data is resampled to a resolution of 0.005° (1/100 of the
resolution of the reforecasts) based on the nearest cell. Secondly, a fishnet which has a resolution of 0.5° and is coincidence
with the grids of the reforecasts is produced. In each grid of the fishnet, there are 10,000 cells of the IMS snow data as the
resolution of the IMS snow data after resampling is one-hundredth of that of the fishnet. The number of cells which are
covered by snow is counted and then divided by 10,000 to get the ratio of the snow cover cells in each grid. Finally, the
50   ratios of the snow cover cells in every grid of the fishnet are calculated to obtain the IMS snow cover fractions with the same
grids of the reforecasts.

Section 3: this section is insufficient; more information is necessary. It should include the method used to assimilate IMS
data (assimilation method, analysed variable, snow model used for background, frequency of assimilation, etc.). Please
55   clarify the role of IFS in your set-up.

Reply: This paper is not a data assimilation paper per se; it relies on previous studies, cited in the paper, to describe the data
assimilation set-up. However, we acknowledge that providing minimum information in the data assimilation approach is
useful for the reader. So, the section was updated to provide more information on the IMS snow cover assimilation method,
60   link to the relevant documentation, and details on the analysis experiments and analyzed variables: The IMS snow data
assimilation method relies on a two-dimensional optimal interpolation approach which is used to analyse the IFS land
surface model (HTESSEL, Balsamo et al., 2009; Dutra et al., 2010) snow depth, with adjustment of snow density when fresh
snow is added by positive increments. Full details on the snow data assimilation method are provided in the IFS
documentation CY45R1, Chapter 9 (IFS CY45R1, 2018).
65

References:

Balsamo, G., Viterbo, P., Beljaars, A., van den Hurk, B., Hirschi, M., Betts, A. K., and Scipal, K.: A Revised Hydrology for the ECMWF Model: Verification from Field Site to Terrestrial Water Storage and Impact in the Integrated Forecast System, Journal of Hydrometeorology, 10, 623-643, https://doi.org/10.1175/2008jhm1068.1, 2009.

70    Dutra, E., Balsamo, G., Viterbo, P., Miranda, P. M. A., Beljaars, A., Schär, C., and Elder, K.: An Improved Snow Scheme for the ECMWF Land Surface Model: Description and Offline Validation, Journal of Hydrometeorology, 11, 899-916, https://doi.org/10.1175/2010jhm1249.1, 2010.

IFS Documentation CY45R1 - Part II : Data assimilation, in: IFS Documentation CY45R1, IFS Documentation, ECMWF, https://doi.org/10.21957/a3ri44ig4, 2018.

75

Line 128: Did you evaluate/compare the two initial states?

Reply: The impact of IMS snow cover assimilation was extensively evaluated in other studies such as Orsolini et al. 2019.

80    Reference: Orsolini, Y., Wegmann, M., Dutra, E., Liu, B., Balsamo, G., Yang, K., de Rosnay, P., Zhu, C., Wang, W., Senan, R., and Arduini, G.: Evaluation of snow depth and snow cover over the Tibetan Plateau in global reanalyses using in situ and satellite remote sensing observations, The Cryosphere, 13, 2221-2239, https://doi.org/10.5194/tc-13-2221-2019, 2019.

Line 158: The reader is not yet aware of an "inherent" model precipitation excess. Please explain and clarify what you mean.
85    Furthermore, IMS only gives information on the presence of snow, not on the state of the snowpack, so this sentence is misleading.

Reply: Thanks for the comments. This sentence has been rewritten: The positive bias in snow depth is also much reduced in the DA reforecasts, which is consistent with the decreases in snow cover fraction due to the added assimilation of IMS snow
90    cover.

Line 173: Could you explain how snow density is affected by IMS data assimilation?

Reply: With the added snow assimilation, the snowfall of the DA reforecasts is more than that of the control reforecasts for
95    the ETP and around the boundary of the WTP and ETP in the southern TP. An increase of snowfall will lead to more (new) low-density snow depositing on the ground. Analysis about snow density has been moved to Supplementary as it is unnecessary to support the main conclusions.

Line 179: Could you explain this statement? How is albedo affected by the assimilation of IMS data?

100

Reply: With the added snow assimilation, the snow cover fraction changes a lot, especially for the ETP and around the boundary of the WTP and ETP. Typically, snow and ice have high reflectivity with albedo values of 0.8 and above, and land has intermediate values between about 0.1 and 0.4. Because of the changes in snow cover fraction, the land surface albedo also changes after snow assimilation. We have modified this sentence in the text as: Since the changes of snow cover leads to changes in land surface albedo after snow assimilation, Figure 5 presents the spatial differences in land surface albedo between the two ensemble reforecasts.

Line 188: Too vague! Which variable are you talking about?

Reply: Sorry for the confusion. This sentence has been rewritten: the main points are that snow assimilation reduces the positive biases of snow cover fraction and snow depth in spring over most areas of the Tibetan Plateau.

Lines 190-191: Reduced in Depth or in Cover Fraction? Please clarify what you mean by that.

Reply: This sentence has been rewritten: The reduced snow cover fraction leads to a diminished surface albedo.

Lines 207-208: How would you explain the decrease in correlation when using DA?

Reply: Thanks for the comment. As the data assimilation is performed for snow variables rather than temperature directly, the decrease in correlations of temperature reforecasts might be attributed to the changes in complex regional thermodynamics processes. Moreover, although the correlations of temperature reforecasts decrease after snow assimilation, the added snow assimilation still makes sense as the temperature biases improve. We have incorporated the explanations into the Discussions section.

Line 210: More information should be given as to what we see in these figures: what are the + signs, explain why the CC decrease with DA, explain the high median in CC in WTP (vs. ETP), etc.

Reply: Thanks for the comments. The + represents the member which pasts the first and third quartiles when calculating the metrics. The reason why the CCs decrease with DA has been explained and incorporated into the Discussions section: As the data assimilation is performed for snow variables rather than temperature directly, the decrease in correlations of temperature reforecasts might be attributed to the changes in complex regional thermodynamics processes. As for the high median in CC in WTP (vs. ETP), the topography is more rugged in the ETP than in the WTP, leading to the large temperature variability in the ETP which makes the temperature simulation more difficult and finally causes the lower correlations in the ETP than in the WTP.

Figures S1-5 cannot be in supplementary material if you are referring to them in the text. Multiple paragraphs refer to and explain these figures and the manuscript cannot be fully understood without direct access to them. Please insert them in the manuscript.

Reply: Thanks for the comment. We have moved the descriptions of Fig. S1-5 to Supplementary. This is because there have been already many figures in the manuscript and the descriptions of Fig. S1-5 are unnecessary to support our main conclusions.

Line 241: Explain why.

Reply: The snow assimilation above 1500 m over the Tibetan Plateau mainly reduces the positive biases in snow cover fraction and snow depth in spring, while in summer, the impact of added snow assimilation on the snowpack state is quite little. Therefore, the changes in 10 m wind are also small in summer. This sentence has been rewritten in the revised manuscript: However, the added snow assimilation has little impact on the 10 m wind field in summer as the snowpack state changes little at the meantime.

Line 255: lacks precision. Is this cumulative and total (i.e., solid + liquid) precipitation? In this case, how do you convert solid precipitation to mm? Is it averaged over the domain? Lots of information missing in the text and in the corresponding figures.

Reply: Sorry for the confusion. The precipitation here refers to daily and total liquid precipitation (rainfall + snowfall) which has been averaged over the domain (i.e., the western and eastern Tibetan Plateau). Actually, the units of the raw precipitation outputted from model are depth in meters of water equivalent. It is the depth the water would have if it were spread evenly over the grid box. We have further explained and described the figures in the revised manuscript.

Line 335+: The discussion about the changes in snowpack states should be discussed before their impact on the atmosphere. Please consider reorganizing the Discussion.

Reply: Thanks for the comment and the Discussions section have been reorganized. The discussion about the changes in snowpack states is firstly presented in the revised manuscript.

Line 345 and throughout the manuscript: please clarify whether you are talking about snow albedo or total land albedo (i.e., snow-free and snow-covered land as well as vegetation)

170     Reply: Sorry for the confusion. The forecast albedo in the manuscript refers to land surface albedo. We have replaced "forecast albedo" with "land surface albedo" throughout the manuscript.

269-271: Could you explain why?

175     Reply: The smaller correlations and larger biases of the precipitation reforecasts after snow assimilation may be partly caused by the uncertainties in observations. The bulk of the precipitation over the TP falls as snow in winter and spring, but the GPM products tend to underestimate snowfall which may result in underestimation of total precipitation. However, the snowfall reforecasts become larger after snow assimilation, especially in the eastern Tibetan Plateau and around the boundary of the western and eastern Tibetan Plateau, which may further lead to the smaller correlations and larger biases

180     between the precipitation reforecasts and GPM precipitation. Relative explanations have been added into the Discussions of revised manuscript.

Lines 390-391: This is already mentioned just above in point 2.

Conclusions: Point 2 should be snow specific. Point 3 should be specific to the impact of snow DA on the atmosphere.

185

Reply: Thanks for the comment. The Conclusions have been rewritten:

(1) The snow cover fraction and snow depth of the two ensemble reforecasts are larger than the observations for most places of the TP. With the snow assimilation, the snow cover fraction and snow depth of the reforecasts are closer to the observations. With snow assimilation, the snow cover fraction and snow depth are less for the ETP and around the boundary

190     of the WTP and ETP than that from the control reforecasts, and the land surface albedo of the DA reforecasts is also smaller than that of the control reforecasts for the regions where the snow cover fraction reduces. However, the snowfall of the DA reforecasts is more than that of the control reforecasts for the ETP and around the boundary of the WTP and ETP in the southern TP.

(2) When using the CN05.1 temperature as benchmark, the two ensemble reforecasts can capture the seasonal tendencies of

195     the observed temperature. However, the reforecasts tend to underestimate daily temperature. The added snow assimilation improves mean error but decreases correlations of the temperature reforecasts when comparing with the CN05.1 data. The temperature of the DA reforecasts is significantly higher than that of the control reforecasts for the ETP and around the boundary of the WTP and ETP due to the decreased snowpack and smaller land surface albedo after snow assimilation. Moreover, the wind (at 10 m) transports more heat from surrounding regions to the centre in the ETP after snow assimilation,

200     which further leads to a higher temperature.

(3) When using the GPM precipitation as benchmark, the precipitation reforecasts perform better in the WTP than in the ETP. With the snow assimilation, the biases between the precipitation reforecasts and GPM precipitation becomes larger in the

ETP while smaller in the WTP, which may be partly because of the uncertainty from the GPM observations. The precipitation of the DA reforecasts is significantly more than that of the control reforecasts for the ETP and around the boundary of the WTP and ETP as the higher temperature in these regions enables more moisture to be carried to the atmosphere. Moreover, most of the increased precipitation is in the form of rainfall.

The English language needs to be improved throughout the manuscript, but more specific corrections are detailed below.

Reply: Thanks for the comments. We have improved the English language throughout the manuscript by native speakers and modified the manuscript and figures according to all the specific corrections.

**Technical corrections:**

Hectopascal units are to be written hPa (and not hpa), please correct throughout the manuscript and figures.

Reply: Thanks for the comment. Hectopascal units have been corrected as hPa throughout the manuscript and figures.

Line 20: replace underestimate by underestimating.

Reply: Done.

Lines 64-66: please improve wording.

Reply: Thanks for the comment. This sentence has been rewritten: However, assimilating the IMS snow data but only below 1500 m elevation might influence the forecasting ability over the TP, and inclusion of IMS above 1500 m elevation are probably beneficial to seasonal forecasts at the regional scale.

Lines 138-139: please improve wording.

Reply: Thank for the comment. This sentence has been rewritten: Considering that the only difference between the twin experiments is whether assimilating IMS above 1500 m over the TP, the snow cover is firstly analysed to evaluate the effects of the snow assimilation.

Lines 195 and 257: replace 'the' 5-day by 'a' 5-day

Reply: Done.

Line 204: is CC defined before?

Reply: Thanks for the comment. The Spearman's correlation coefficient (CC) is now defined the first time it is used in the text, i.e., in section 4.2.1 "Evaluation of the temperature reforecasts".

Line 285: please improve wording.

Reply: Thanks for the comment. This sentence is unnecessary and has been removed.

Line 332: "perform not well", please rephrase.

Reply: "the GPM products usually perform not well in detecting snowfall" has been replaced with "the GPM products tend to underestimate snowfall".

Line 362: "leads to the more precipitation"

Reply: This sentence has been removed since the Discussions have been reorganized.

Line 388: the use of smaller and larger in this sentence is incorrect, please rephrase.

Reply: This sentence has been removed since the Conclusions have been reorganized. We have carefully checked it throughout the manuscript.

---

## Author Comment (AC2)

**Reply to Referee comment 2**

Dear Editor and Reviewers:

We would like to thank the editor and all reviewers for their valuable suggestions and comments on the manuscript. These comments have not only improved the quality of the current manuscript, but also are beneficial to our future research in general. All point-by-point responses are presented as follows and we have carefully revised the manuscript based on these comments. For clarity, all comments are given in the original version, while responses are marked in blue.

**General comments:**

The omission of snow cover assimilation over the Tibetan Plateau in the ECMWF forecast system gives the authors a compelling experiment (had ECMWF really not already done that experiment in making the decision not to assimilate snow cover above 1500 m elevation?). There are few surprises in the results once we have learnt that the assimilation decreases spring snow cover, but it is still worth seeing the results.

Reply: Thanks for the positive evaluations and comments, all the comments and corrections have been addressed and incorporated into the revised manuscript. Moreover, IMS snow cover assimilation in mountainous areas is constantly evaluated to address the complex feedback between the surface and the atmosphere. Moving towards coupled assimilation ECMWF aims at enhancing the consistency between the different Earth system components, which will allow better exploitation of observations which are sensitive to the surface (such as snow cover).

**Corrections:**

Line 15: It would be more informative to say that IMS snow data are assimilated, but only below 1500 m elevation.

Reply: Thanks for the comment. This sentence has been rewritten: However, the SEAS5 is produced with assimilating the Interactive Multisensor Snow and Ice Mapping System (IMS) snow data, but only below 1500 m elevation which may affect the forecasting ability of SEAS5 over the region.

Line 20: "while underestimating"

Reply: Sorry for the mistake. This grammar mistake has been corrected.

Line 57: A statement of why IMS is not assimilated in IFS and hence SEAS5 is required. I think that this is not clear either here or in de Rosnay et al. (2012).

35   Reply: IMS snow cover assimilation improves snow and surface representation, however it has a complex impact on the atmospheric forecasts. IMS snow cover assimilation in mountainous areas is constantly evaluated to address the complex feedback between the surface and the atmosphere. Moving towards coupled assimilation ECMWF aims at enhancing the consistency between the different Earth system components, which will allow better exploitation of observations which are sensitive to the surface (such as snow cover).

40

Line 70: State the resolution in terms that will be comprehensible to general readers.

Reply: Thanks for the comment. The resolution has been stated in more clear terms: The configuration for these experiments was largely similar to the current SEAS5 but with lower atmospheric (~ 0.44°) and ocean (~ 1°) resolution and a newer IFS

45   model cycle (CY45R1).

Line 95: Why is "zonal statistics" stated here? It is not just zonal averaging that is required to go from 4 km to 0.5-degree resolution.

50   Reply: Sorry for the confusion. The method used to obtain the 0.5° IMS snow cover fraction from 4-km IMS snow cover data has been further explained in the revised manuscript: The raw IMS snow data used in this study has a resolution of 4 km while the resolution of the reforecasts is 0.5°. The raw IMS snow data is post-processed as following steps to get the IMS snow cover fractions with the same grids of the reforecasts. Firstly, the raw IMS snow data is resampled to a resolution of 0.005° (1/100 of the resolution of the reforecasts) based on the nearest cell. Secondly, a fishnet which has a resolution of 0.5°

55   and is coincidence with the grids of the reforecasts is produced. In each grid of the fishnet, there are 10,000 cells of the IMS snow data as the resolution of the IMS snow data after resampling is one-hundredth of that of the fishnet. The number of cells which are covered by snow is counted and then divided by 10,000 to get the ratio of the snow cover cells in each grid. Finally, the ratios of the snow cover cells in every grid of the fishnet are calculated to obtain the IMS snow cover fractions with the same grids of the reforecasts.

60

Section 4.1: Because only the TP is shown in Figure 2, "of the TP" does not need to be stated so many times.

Reply: We have deleted it.

65   Line 169: "(the DA reforecasts minus the control reforecasts)" has already appeared in this sentence.

Reply: Done.

Line 179: I don't think that snow albedo depends directly on density in IFS. The differences in albedo are clearly dominated by differences in SCF.

Reply: Sorry for the misleading. This sentence has been modified as: Since the changes of snow cover leads to changes in land surface albedo after snow assimilation, Figure 5 presents the spatial differences in land surface albedo between the two ensemble reforecasts.

Line 204: CCs used here but not explained until line 266.

Reply: The explanation of CCs has been added in Line 209 and removed from Line 273.

Line 275: The supplementary figures should either be in the main paper (which will result in there being a lot of figures) or the discussion of them should be in the supplement.

Reply: Thanks for the comment. The discussions of Fig. S1-5 have been moved to Supplementary.

Line 299: "for the two reforecasts"

Reply: Done.

A lot of the Discussion section simply restates results that will be stated again in the Conclusions.

Reply: Thanks for the comment. The Discussions section has been reorganized and condensed.

Line 440: de Rosnay et al. (2012) reference appears twice. de Rosnay et al. (2014) is missing.

Reply: Sorry for the mistakes. de Rosnay et al. (2012) has been corrected to be de Rosnay et al. (2014).

Figure 1 shows elevation; DEM is just how elevation is specified. And even common acronyms should be explained.

Reply: "DEM" in Fig. 1 has been replaced with "Elevation", and the Figure 1 caption has been rewritten: The location and elevation of the Tibetan Plateau (TP) and the location of climate observation stations.

The Figure 2 caption should state that the difference between reforecasts is with – without.

Reply: The Figure 2 caption has been modified as: (d-f) The spatial differences in snow cover fraction between the two reforecasts (with – without snow assimilation).

The description of columns in Figures 3, 8 and 13 is actually rows.

Reply: Done.

---

## Author Comment (AC3)

**Reply to Referee comment 3**

Dear Editor and Reviewers:

We would like to thank the editor and all reviewers for their valuable suggestions and comments on the manuscript. These comments have not only improved the quality of the current manuscript, but also are beneficial to our future research in general. All point-by-point responses are presented as follows and we have carefully revised the manuscript based on these comments. For clarity, all comments are given in the original version, while responses are marked in blue.

This work describes the impact of the usage of snow cover data assimilation above 1500m in the ECMWF seasonal prediction system over the Tibetan Plateau, by means of analyzing two set of reforecast initialised in different ways. One set of reforecasts is initialised with an analysis using IMS snow cover observations for z>1500m in the snow data assimilation system. The other set was initialised without using these observations above the z=1500m orography threshold.

The experimental setup gives the opportunity to study the impact of snow initial conditions on seasonal time-scales over the Tibetan plateau. Results are interesting and worth publishing. However, I have some comments that I would like the authors to address before the work is published, that are reported below. For these reasons, I suggest major revisions on the manuscript.

Reply: Thanks for the positive evaluations and comments, all the comments and suggestions have been addressed and incorporated into the revised manuscript.

**General comments:**

I found the description of the model setup not clear enough, missing important details or reported in a confusing way. I suggest the authors to reorganise the "Data" and "Methods" sections. All information regarding the model (horizontal resolution, number of vertical levels in atmospheric and ocean model etc.) should be reported in one section and meaning of acronyms clearly explained (e.g. Ln. 70, "ORCA1_Z75"). For instance, Ln. 90 says that "The reforecasts have a spatial resolution of 0.5°"; however, Ln. 70 and Ln. 120 says that a TCo grid is used.

Reply: Sorry for the confusions. The "Data" and "Methods" sections have been reorganized as the "Methods and Data" section. All information regarding the model is reported in section 3.1 "Methods". More details on the model setup will be added into the manuscript. The resolutions of atmospheric and ocean model have been stated in more clear terms: The configuration for these experiments was largely similar to the current SEAS5 but with lower atmospheric (~ 0.44°) and ocean (~ 1°) resolution and a newer IFS model cycle (CY45R1).

The Results section is in many places descriptive and can be shortened, improving conciseness and clarity. For instance, the discussion of scores (CC, MAE) in 4.2.1 and 4.3 could be simplified. Another example is the discussion about snow density, which is not linked with the other variables; the underlying physical mechanism for which density is lower in the DA experiment is not clear from the text. Also, there is large usage of "supplementary" figures, in particular in Sect. 4.3, two paragraphs of discussion of "Supplementary" material. If those figures are important for the discussion maybe the authors can think of moving those in the main text? Otherwise, I would suggest the authors to rearrange the text, moving for instance details that are unnecessary to support the main conclusions to an appendix?

Reply: Thanks for the comments. The Results section has been simplified to improve conciseness and clarity, especially for the discussion of scores in 4.2.1 and 4.3. The discussion about snow density has been moved to the Supplementary as it is not linked with the other variables. The inconsistent change between snow density and forecast albedo is because that the variable "forecast albedo" in the manuscript is referred to land surface albedo rather than snow albedo. Moreover, we have rearranged the text, moving Fig. S1-5 to the Supplementary as they are unnecessary to support the main conclusions.

I acknowledge that the proposed methodology was developed with the climatology differences between West and East Tibetan plateau in mind. However, it looks to me that the main differences in precipitation (Fig. 13), or snow depth (Fig.3), are in a south-located region on the edge of the (arbitrary?) 95° line chosen by the authors. How results are sensitive to the choice of this longitude value?

Reply: Thanks for the comments. The longitude 95°E is used as the boundary of west and east Tibetan plateau (TP) in this study considering the high spatial variability of precipitation and temperature in the TP. The choice of the longitude 95°E is not subjective but according to the climate pattern in the TP, i.e., the climate pattern in the east TP (> 95°E) is usually considered as wet, while it is usually considered as dry in the west TP (< 95°E), and also refers to previous studies (Qian, et al., 2003; Li et al., 2020). Moreover, the only difference in the two reforecasts is whether assimilating IMS snow cover data above 1500 m. The significant differences in snow cover fraction before and after snow assimilation are concentrated around the longitude 95°E in the southeastern TP, leading to the significant changes in precipitation and snow depth at regional scale.

Reference:
Qian, Y. F., Zheng, Y. Q., Zhang, Y., and Miao, M. Q.: Responses of China's summer monsoon climate to snow anomaly over the Tibetan Plateau, International Journal of Climatology, 23, 593-613, 10.1002/joc.901, 2003.
Li, D., Yang, K., Tang, W., Li, X., Zhou, X., and Guo, D.: Characterizing precipitation in high altitudes of the western Tibetan plateau with a focus on major glacier areas, International Journal of Climatology, 40, 5114-5127, https://doi.org/10.1002/joc.6509, 2020.

**Specific comments:**

Abstract: I would make it clearer that reforecasts are initialised with analysis produced with/without snow assimilation above z=1500m.

Reply: Thanks for the comments. We have made this clear in Line 16-18 in Abstract now: To investigate the impacts of snow assimilation on the forecasting of snow, temperature and precipitation, twin ensemble reforecasts are initialized with and without snow assimilation above 1500 m elevation over the Tibetan Plateau for the spring and summer 2018.

I found the last sentence of the abstract rather vague. Can you be more specific, e.g. which component of the surface energy balance? A plot showing which surface flux is mostly affected would be important to support this last statement.

Reply: Sorry for the confusions. We admit that this statement is unsuitable as the surface energy balance is not the focus of this study. This sentence has been rewritten: Overall, the snow assimilation can improve the seasonal forecasts through the interaction between land and atmosphere.

Ln 70: I think more details on the model setup should be provided for people not familiar with the specific model (see main comments).

Reply: Thanks for the comments. More details on the model setup will be added into the manuscript.

From the "Methods" section, is not clear if the dedicated analysis experiments are land surface analysis only or include the analysis of the entire atmosphere + land. Please clarify in the text.

Reply: Sorry for the confusion. We have clarified that the dedicated analysis experiments include the analysis of the atmosphere and land in Line 111-113: In this study, we analysed the impacts of snow assimilation over the TP on the snowpack state (snow cover fraction, snow depth and snowfall) as well as on near surface variables (land surface albedo, 2m air temperature, 10m wind and total liquid precipitation) and upper air variable (geopotential height and temperature at 600 hPa).

Was the orography threshold for using IMS observations in the snow assimilation system only removed for the Tibetan plateau region? Or was removed globally, and then the analysis focused on the Tibetan plateau region?

100 Reply: The orography threshold for using IMS observations in the snow assimilation system was removed specifically on the Tibetan Plateau region and maintained elsewhere. The analysis focused on the Tibetan plateau region. We have added the clarification in the revised manuscript.

For how long the dedicated analysis were run? Are there possible model spin-ups in the land or atmospheric fields that
105 should be taken into account?

Reply: There are enough model spin-ups in the land and atmospheric fields. We will add relative information in the revised manuscript.

110 Ln 159: What does "inherent" means? Also snow model biases can contribute to snow depth errors.

Reply: Sorry for the confusion. This sentence has been rewritten in the revised manuscript: The positive bias in snow depth is also much reduced in the DA reforecasts, which is consistent with the decreases in snow cover fraction due to the added assimilation of IMS snow cover.
115
Sect 4.1: The mechanism linking the change in snowfall, snow density and albedo is not clear from this section. An increase of snowfall in the forecast would be associated with more (new) low-density snow depositing on the ground. A fresher snowpack would be associated with a higher albedo. However, the authors found that the albedo decreases in the DA simulation. Why?
120
Reply: Thanks for the comments. We agree with that an increase of snowfall in the forecast would be associated with more (new) low-density snow depositing on the ground which has been presented in the manuscript, and a fresher snowpack would be associated with a higher snow albedo. However, the forecast albedo used in the analysis refers to the land surface albedo rather than the snow albedo. Due to the smaller snow cover fraction after the added snow assimilation, the land
125 surface albedo decreases accordingly. The "forecast albedo" has been replaced with the "land surface albedo" throughout the manuscript to avoid confusion.

1: it would be useful to have an indication on where the data assimilation is acting, that is, highlighting the grid points with orography > 1500m (from the current figure it is hard to see).
130
Reply: Thanks for the comment. We acknowledge that it would be useful to have an indication on where the data assimilation is acting. However, regions where the orography > 1500 m account for 98.7% of the whole study area (Fig. R1). Considering that there are 1013 grid points in total, fewer than 13 grid points are located in regions where the orography <=

1500 m, which are mainly distributed in the southern boundary of the study area and have few impacts on the analyses and conclusions. Highlighting the grid points with orography > 1500m might be unnecessary in this situation. Moreover, we have tried to plot the boundary of the regions with orography > 1500 m, however, it is almost coincident with the boundary of the study area. We have modified Line 75-76 in the text of revised manuscript to state that almost all of the study area is influenced by the added snow assimilation: Regions where the orography > 1500 m account for 98.7% of the whole study area.

[Figure]

**Figure R1: The location and elevation of the Tibetan Plateau (TP) and the location of climate observation stations.**

Ln 204-213: Does the fact that CC is lower in the reforecast with snow DA mean that the temperature variability is worsened, but the temperature biases compared to CN05.1 are improved (as shown by the reduced MAE)?

Reply: We admit that the correlations of temperature reforecasts decrease after snow assimilation. As the data assimilation is performed for snow variables rather than temperature directly, the decrease in correlations of temperature reforecasts might be attributed to the changes in complex regional thermodynamics processes. Moreover, although the correlations of temperature reforecasts decrease after snow assimilation, the added snow assimilation still makes sense as the temperature biases improve.

Ln 233: How would you explain that the correlation against in situ observation gives a different result than the correlation against CN05.1 product?

155

Reply: The correlations against in situ observations presented in Fig. S2 (now Fig. S3) are spatial correlations, while those against CN05.1 product presented in Fig. 7 are temporal correlations. After calculating the temporal correlations between the two ensemble reforecasts and in situ observations, the results are similar with the temporal correlations against CN05.1 product. We have further clarified in the revised manuscript that the correlations against CN05.1 and GPM products are

160 temporal correlations.

[Figure]

**Figure R2: The temporal correlations and mean absolute error between the temperature reforecasts and in-situ observations.**

165 Ln 239: It is not clear what "obvious" mean here. Please rephrase. See also at ln 358.

Reply: Thanks for the comment. Line 238-240 (in the raw manuscript) has been rewritten: With snow assimilation, the wind speed of the DA reforecasts is much larger than that of the control reforecasts in the eastern Tibetan Plateau in either the spring or the whole period. Line 358 (in the raw manuscript) has been rewritten: Therefore, the 10 m wind field is also

170 analysed and the centre of changes in 10 m wind field is observed in the ETP, which is coincident with the centre of changes in snow and temperature in the ETP.

Ln 243: what is the (mean) height above the Tibet plateau of the 600hPa surface?

175 Reply: We will further check the (mean) height above the Tibet plateau of the 600 hPa surface.

Ln 266: The Spearman's correlation coefficients (CCs) should be defined the first time it is used in the text.

Reply: Done.

Ln 324: it improves in mean error but decrease correlation.

Reply: This sentence has been written: The snow assimilation improves mean error but decreases correlations of the temperature reforecasts when comparing with the CN05.1 data.

I found the argument of "horizontal heat transport" a bit speculative. The authors should also show horizontal temperature maps to clearly see if warmer air is advected with the wind. For instance, could the "convergence zone" cause colder temperature from surrounding snow area (or higher mountains) to be advected over the region?

Reply: Thanks for the comment. Fig. R3 presents the spatial differences in temperature at 600 hPa between the two reforecasts. It can be seen that in spring and the whole period, the temperature at 600 hPa of the DA reforecasts is higher than that of the control reforecasts for most areas of the TP, especially for the ETP and around the boundary of the WTP and ETP. The spatial differences in temperature at 600 hPa are similar with those in geopotential height at 600 hPa but with reversed changes, i.e., the temperature increases when the geopotential height decreases. Furthermore, the increases in temperature are also consistent with the increases in wind, as the warmer air is advected with the wind. We have added the horizontal temperature maps and relative explanations into the revised manuscript.

[Figure]

**Figure R3: The spatial differences in temperature at 600 hpa (°C) between the two ensemble reforecasts. The stippled regions show the statistical significance of the differences identified by the t-test at a 5% significance level.**

It would be useful to have a time series of snow depth, similarly to what it is provided for air temperature and precipitation. It would enable understanding if the increased snowfall in the snow DA reforecast compensates to some extent differences due to the initialisation. It would also clarify differences in summertime snow melt in the two reforecasts.

205

Reply: Thanks for the comment. The time series of snow depth from April 1st to July 31st for the two ensemble reforecasts and TPSD are presented in Fig. R4. The snow depth was averaged over the domain (i.e., the WTP and ETP) and the times series were smoothed by a 5-day moving windows. The blue area and line represent the ranges and ensemble-mean of the control reforecasts, respectively; while the orange area and line represent the ranges and ensemble-mean of the DA

210 reforecasts, respectively. The black line represents TPSD data. Both in the WTP and ETP, the ensemble-means of the snow depth of the two reforecasts are higher than those of the TPSD data. However, the snow depth of the DA reforecasts is closer to the TPSD data than that of the control reforecasts. The differences in snow depth between the two reforecasts decrease with time. In the WTP (Fig. R4a), the snow depth of the control reforecasts is higher than that of the DA reforecast for the whole period, while in the ETP, the snow depth of the two ensemble reforecasts is almost the same in the summer. Although

215 the snow depth of the two ensemble reforecasts has an overall downward trend, the snow depth of the DA reforecasts increases around April 15th, which might be contributed to the increases in snowfall in spring after added snow assimilation. The descriptions about the time series of snow depth have been added into the revised manuscript.

[Figure]

**Figure R4: The time series of snow depth averaged over the domain from April 1st to July 31st for the two ensemble reforecasts and TPSD data in the (a) west Tibetan Plateau and (b) east Tibetan Plateau.**

**Technical comments:**

Ln 121: typo, "OCEAN5".

Reply: Done.

Ln 204: "CC" is not defined in the text.

230 Reply: The Spearman's correlation coefficient (CC) is now defined the first time it is used in the text, i.e., in section 4.2.1 "Evaluation of the temperature reforecasts".

Ln 269: suggestion: I would say "lower", not "weak".

235 Reply: Done.

Fig. 13 legend looks wrong to me. Is it not the "spatial differences in daily precipitation (mm) between the ensemble reforecasts and GPM data" the top and middle row (not column)?

240 Reply: Sorry for the mistake. The caption of Fig. 13 has been corrected: The spatial differences in daily precipitation (mm) between the ensemble reforecasts and GPM data (top and middle rows), and between the two reforecasts (bottom row). The stippled regions show the statistical significance of the differences identified by the t-test at a 5% significance level. The same mistakes in Fig. 3 and Fig. 8 have also been corrected.

---

## Author Response (AR2)

**Reply to Editor decision**

**Comments to the author:**

Your revised manuscript was returned to Reviewers 1 and 3, who have now both recommended publication. There is one

5    remaining to issue to address in Section 4.2.2: as noted by Reviewer 3, can you please consider comment regarding horizontal heat transport?

This is the final clarification, after which the manuscript will proceed to publication. Thanks very much for your contribution to The Cryosphere.

10    Dear editor:

We would like to appreciate the editor and all reviewers for their valuable suggestions and comments on the manuscript. These suggestions and comments have further improved the quality of the manuscript. We have carefully checked all the comments and all point-by-point responses are presented as follows. We have also carefully revised the manuscript based on

15    these comments.

Best regards,

Wei Li and co-authors

**Reply to Referee comment 1**

**Comments:**

The manuscript is now much improved and the authors appropriately responded to all my comments. I have only one technical correction to point out. Line 220 of the revised manuscript with track-change: the wording "which pasts" is incorrect, please rephrase.

Reply: Thanks for the positive evaluations. Line 220 of the revised manuscript has been modified as: The + in the figure represents the outlier when calculating the metrics.

**Reply to Referee comment 3**

**General comments:**

The authors have responded to most of my questions in a satisfactory way and improved the text accordingly where needed, e.g. the "Methods and Data" Section. After the reading of the revised manuscript, I have only a few minor remarks left, which I'd encourage the authors to consider, in particular the one on the horizontal heat transport, before the paper is published.

Reply: Thanks for the positive evaluations and valuable comments on the manuscript. These comments have further improved the quality of the manuscript. All point-by-point responses are presented as follows and we have carefully revised the manuscript based on these comments. For clarity, all comments are given in the original version, while responses are marked in blue.

**Specific comments:**

Ln 119 – 129: Not sure what do you mean by "fishnet". Is it not simply a grid with same horizontal resolution of the reforecasts, allowing the conversion from a high resolution binary snow cover information to a low resolution fractional snow cover, as you have described later? Also ln 121-122 could be reworded, to make it clear that the goal of this post-processing steps is to convert a binary higher-resolution information to a lower resolution fractional information.

Reply: Thanks for the comments. Line 119-128 has been modified as: Since IMS snow data was assimilated in the twin analysis experiments, the performance of IMS snow data was evaluated. The IMS snow data used in this study was retrieved from the National Snow and Ice Data Centre (NSIDC) and has a resolution of 4 km. More details about this dataset can be found in https://nsidc.org/data/g02156. In this study, the high resolution binary IMS snow data was post-processed as following steps to get the lower resolution fractional IMS snow cover. Firstly, the raw IMS snow data was resampled to a resolution of 0.005° (1/100 of the resolution of the reforecasts) based on the nearest cell. Secondly, a grid with same horizontal resolution of the reforecasts was produced. In each cell of the grid, there were 10,000 pixels of the IMS snow data as the resolution of the IMS snow data after resampling was one-hundredth of that of the cell. The number of pixels which were covered by snow was counted and then divided by 10,000 to get the ratio of the snow-covered pixels in each cell. Finally, the ratios of the snow-covered pixels in every cell of the grid were calculated to obtain the IMS snow cover fraction with same horizontal resolution of the reforecasts.

Ln 171: It seems that "Fig. R5a" is not the correct figure caption?

Reply: Sorry for the mistake. The figure caption in Line 171 has been corrected as Fig. 3a.

65  Ln 273: Not sure which "aforementioned model excess in precipitation" studies the authors are referring to here. Maybe for a reader would be better the Section or studies to be explicitly mentioned. On precipitation biases (in reanalyses system), worth mentioning Orsolini et al. 2019.

Reply: Thanks for the comments. Line 272-274 has now been rewritten as: Moreover, the ensemble-mean precipitation of
70  the two reforecasts is much more than GPM precipitation before June 25th, in line with the excess precipitation in reanalyses system and climate and forecast models which has been mentioned in Orsolini et al. (2019) and Su et al. (2013).

References:
Orsolini, Y., Wegmann, M., Dutra, E., Liu, B., Balsamo, G., Yang, K., de Rosnay, P., Zhu, C., Wang, W., Senan, R., and
75  Arduini, G.: Evaluation of snow depth and snow cover over the Tibetan Plateau in global reanalyses using in situ and satellite remote sensing observations, The Cryosphere, 13, 2221-2239, https://doi.org/10.5194/tc-13-2221-2019, 2019.
Su, F., Duan, X., Chen, D., Hao, Z., and Cuo, L.: Evaluation of the Global Climate Models in the CMIP5 over the Tibetan Plateau, Journal of Climate, 26, 3187-3208, https://doi.org/10.1175/jcli-d-12-00321.1, 2013.

80  Section 4.2.2, ln 253 – 258: I am still not fully convinced about the horizontal heat transport argument as presented by the authors. Bottom row of Fig. 10, Spring panel, shows a quite localised temperature change where the snow DA is most active. If horizontal temperature advection plays a major role, I would have expected a more spatially extended signal, due to the horizontal transport of warmer air from nearby regions. Can just be a local signal in surface temperature, extending throughout the lower troposphere, triggering the convergence and therefore higher wind speed? Also, it might be obvious,
85  but could the author better explain the following sentence "the increases in temperature are also consistent with the increases in wind, as the warmer air is advected with the wind" (ln. 256-257)?

Reply: Following the reviewer's comment, we have modified our interpretation of Fig. 10: the temperature change attributed to the effects of a more realistic, decreased snowpack after snow DA extends throughout the lower troposphere (as seen at
90  600hPa), implying convergence and ascent, and low pressure. This convergence is consistent with the increase in wind speed. We have modified or deleted relative descriptions throughout the manuscript, especially in Section 4.2.2, Discussions and Conclusions. Because of the modification, Line 256-257 mentioned by the reviewer have now been revised as: Hence, the local signal in surface temperature extends throughout the lower troposphere. The spatial differences in temperature at 600 hPa are similar with those in geopotential height at 600 hPa but reversed, i.e., the temperature increases when the
95  geopotential height decreases. This increase in temperature implies convergence and ascent. The low pressure and convergence are consistent with the increase in horizontal wind speed (Fig 9).

Discussion and Conclusions: An important point that I think is worth mentioning and couldn't really find there is the feedback of the increased snowfall precipitation in the DA-experiment on temperatures. The increase in snowfall before June leads to an increase in snow depth both in the west and east TP (from fig.3). This can partly explain why the two reforecasts ~ converge in the average daily temperature time series after a month or so. If what I am describing is correct, improving the feedback of snow assimilation on precipitation seems crucial to get all the benefits of improved land-surface initial conditions.

Reply: Thanks for the comments. We have added relative discussions in Line 306-308: However, the snowfall of the DA reforecasts is larger than that of the control reforecasts for the ETP and around the boundary of the WTP and ETP in the southern TP, leading to an increase in the time series of snow depth in spring both in the WTP and ETP for the DA experiment, in Line 320-321: The two reforecasts converge in the average daily temperature time series after a month or so, possibly resulting from the additional snowfall and cooling in the DA reforecasts, and in Line 362-364 in Conclusions: Given its feedback on snowfall and snow depth, it appears important for the forecast models to capture the effect of land-atmosphere interaction upon precipitation to get all the benefits of improved land-surface initial conditions.